# Normed Spaces for Graph Embedding

**Diaaeldin Taha**[*]                                                        *diaaeldin.taha@mis.mpg.de*
*Max Planck Institute for Mathematics in the Sciences,*
*Leipzig, Germany*

**Wei Zhao**[*]                                                                    *wei.zhao@abdn.ac.uk*
*University of Aberdeen,*
*Aberdeen, United Kingdom*

**J. Maxwell Riestenberg**                                          *max.riestenberg@mis.mpg.de*
*Max Planck Institute for Mathematics in the Sciences,*
*Leipzig, Germany*

**Michael Strube**                                                        *michael.strube@h-its.org*
*Heidelberg Institute for Theoretical Studies,*
*Heidelberg, Germany*

**Reviewed on OpenReview:** *https://openreview.net/forum?id=4E2XLydJiv*

## Abstract

Theoretical results from discrete geometry suggest that normed spaces can abstractly embed finite metric spaces with surprisingly low theoretical bounds on distortion in low dimensions. Inspired by this theoretical insight, we highlight in this paper normed spaces as a more flexible and computationally efficient alternative to several popular Riemannian manifolds for learning graph embeddings. Normed space embeddings significantly outperform several popular manifolds on a large range of synthetic and real-world graph reconstruction benchmark datasets while requiring significantly fewer computational resources. We also empirically verify the superiority of normed space embeddings on growing families of graphs associated with negative, zero, and positive curvature, further reinforcing the flexibility of normed spaces in capturing diverse graph structures as graph sizes increase. Lastly, we demonstrate the utility of normed space embeddings on two applied graph embedding tasks, namely, link prediction and recommender systems. Our work highlights the potential of normed spaces for geometric graph representation learning, raises new research questions, and offers a valuable tool for experimental mathematics in the field of finite metric space embeddings. We make our code and data publically available [1].

## 1 Introduction

Graph representation learning aims to embed real-world graph data into ambient spaces while sufficiently preserving the geometric and statistical graph structures for subsequent downstream tasks and analysis. Graph data in many domains exhibit non-Euclidean features, making Euclidean embedding spaces an unfit choice. Motivated by the manifold hypothesis (see, e.g., Bengio et al. (2013)), recent research work has proposed embedding graphs into Riemannian manifolds (Chamberlain et al., 2017; Defferrard et al., 2020; Grattarola et al., 2020; Gu et al., 2019; Tifrea et al., 2019). These manifolds introduce inductive biases, such as symmetry and curvature, that can match the underlying graph properties, thereby enhancing the quality of the embeddings. For instance, Chamberlain et al. (2017) and Defferrard et al. (2020) proposed embedding graphs into hyperbolic and spherical spaces, with the choice determined by the graph structures.

---

[1] https://github.com/andyweizhao/graphs-normed-spaces
 [*] These authors contributed equally to this work.

More recently, López et al. (López et al., 2021; López et al., 2021) proposed Riemannian symmetric spaces as a framework that unifies many Riemannian manifolds previously considered for representation learning. They also highlighted the Siegel and SPD symmetric spaces, whose geometries combine the sought-for inductive biases of many manifolds. However, operations in these non-Euclidean spaces are computationally demanding and technically challenging, making them impractical for embedding large graphs.

In this work, we highlight normed spaces, particularly $\ell_1^d$ and $\ell_\infty^d$, as a more flexible, more computationally efficient, and less technically challenging alternative to several popular Riemannian manifolds for learning graph embeddings. In particular, normed spaces are empirically observed to be easier to train and to perform better in general graph embedding settings that leverage gradient descent. Our proposal is motivated by theoretical results from discrete geometry, which suggest that normed spaces can abstractly embed finite metric spaces with surprisingly low theoretical bounds on distortion in low dimensions. This is evident in the work of Bourgain (1985); Johnson & Lindenstrauss (1984) and Johnson et al. (1987).

We evaluate the representational capacity of normed spaces on synthetic and real-world benchmark graph datasets through a graph reconstruction task. Our empirical results corroborate the theoretical motivation; as observed in our experiments, diverse classes of graphs with varying structures can be embedded in low-dimensional normed spaces with low average distortion. Second, we find that normed spaces consistently outperform Euclidean spaces, hyperbolic spaces, Cartesian products of these spaces, Siegel spaces, and spaces of SPD matrices across test setups. Further empirical analysis shows that the embedding capacity of normed spaces remains robust across varying graph curvatures and with increasing graph sizes. Moreover, the computational resource requirements for normed spaces grow much slower than other Riemannian manifold alternatives as the graph size increases. Lastly, we showcase the versatility of normed spaces in two applied graph embedding tasks, namely, link prediction and recommender systems, with the $\ell_1$ normed space surpassing the baseline spaces.

As the field increasingly shifts towards technically challenging geometric methods, our work underscores the untapped potential of simpler geometric techniques. As demonstrated by our experiments, normed spaces set a compelling baseline for future work in geometric representation learning.

## 2 Related Work

Graph embeddings are mappings of discrete graphs into continuous spaces, commonly used as substitutes for the graphs in machine learning pipelines. There are numerous approaches for producing graph embeddings, and we highlight some representative examples: (1) Matrix factorization methods (Belkin & Niyogi, 2001; Cai et al., 2010; Tang & Liu, 2011) which decompose adjacency or Laplacian matrices into smaller matrices, providing robust mathematical vector representations of nodes; (2) Graph neural networks (GNNs) (Kipf & Welling, 2017; Veličković et al., 2018; Chami et al., 2019a) which use message-passing to aggregate node information, effectively capturing local and global statistical graph structures; (3) Autoencoder approaches (Kipf & Welling, 2016; Salha et al., 2019) which involve a two-step process of encoding and decoding to generate graph embeddings; (4) Random walk approaches (Perozzi et al., 2014; Grover & Leskovec, 2016; Kriege, 2022) which simulate random walks on the graph, capturing node proximity in the embedding space through co-occurrence probabilities; and (5) Geometric approaches (Gu et al., 2019; López et al., 2021) which leverage the geometric inductive bias of embedding spaces to align with the inherent graph structures, typically aiming to learn approximate isometric embeddings of the graphs in the embedding spaces. We note that these categories are not mutually exclusive. For instance, matrix factorization can be seen as a linear autoencoder approach, and geometric approaches can be combined with graph neural networks. Here we follow previous work (Gu et al., 2019; López et al., 2021; López et al., 2021; Giovanni et al., 2022) and use a geometric approach to produce graph embeddings in normed spaces.

Recently, there has been a growing interest in geometric deep learning, especially in the use of Riemannian manifolds for graph embeddings. Those manifolds include hyperbolic spaces (Chamberlain et al., 2017; Ganea et al., 2018; Nickel & Kiela, 2018; López et al., 2019), spherical spaces (Meng et al., 2019; Defferrard et al., 2020), combinations thereof (Bachmann et al., 2020; Grattarola et al., 2020; Law & Stam, 2020), Cartesian products of spaces (Gu et al., 2019; Tifrea et al., 2019), Grassmannian manifolds (Huang et al., 2018), spaces of symmetric positive definite matrices (SPD) (Huang & Gool, 2017; Cruceru et al., 2020),

and Siegel spaces (López et al., 2021). All these spaces are special cases of Riemannian symmetric spaces, also known as *homogeneous spaces*. Non-homogeneous spaces, such as Giovanni et al. (2022), have been explored for embedding heterogeneous graphs. Other examples of mathematical spaces include Hilbert spaces (Sriperumbudur et al., 2010; Herath et al., 2017), Lie groups (Falorsi et al., 2018) (such as the torus (Ebisu & Ichise, 2018)), non-abelian groups (Yang et al., 2020) and pseudo-Riemannian manifolds of constant nonzero curvature (Law & Stam, 2020). These spaces introduce inductive biases that align with critical graph features. For instance, hyperbolic spaces, known for embedding infinite trees with arbitrarily low distortion (Sarkar, 2012), are particularly suitable for hierarchical data. Though the computations involved in working with these spaces are tractable, they incur non-trivial computational costs and often pose technical challenges. In contrast, normed spaces, which we focus on in this work, avoid these complications.

## 3 Theoretical Inspiration

In discrete geometry, abstract embeddings of finite metric spaces into normed spaces, which are characterized by low theoretical distortion bounds, have long been studied. Here we review some of the existence results that motivated our work. These results provide a rationale for using normed spaces to embed various graph types, much like hyperbolic spaces are often matched with hierarchical graph structures. While these results offer a strong motivation for our experiments, we emphasize that these theoretical insights do not immediately translate to or predict our empirical results. Theoretical results and embedding spaces investigated in this work are summarized in Tables 7 and 8 (Appendix).

It is a well-known fact that any $n$-pointed metric space can theoretically be isometrically embedded into $\ell_\infty^n$. For many classes of graphs, the theoretical bound on dimension can be substantially lowered: the complete graph $K_n$ can theoretically be isometrically embeded in $l_1^{\lceil \log_2(n) \rceil}$, every tree $T$ with $n$ vertices can theoretically be isometrically embedded in $\ell_\infty^{\mathcal{O}(\log n)}$, and every tree $T$ with $\ell$ leaves can theoretically be isometrically embedded in $\ell_\infty^{\mathcal{O}(\log \ell)}$ (Linial et al., 1995).

Bourgain showed that similar dimension bounds can be obtained for finite metric spaces in general by relaxing the requirement that the embedding is isometric. A map $f : X \to Y$ between two metric spaces $(X, d_X)$ and $(Y, d_Y)$ is called a *D-embedding* for a real number $D \geq 1$ if there exists a number $r > 0$ such that for all $x_1, x_2 \in X$,

$$r \cdot d_X(x_1, x_2) \leq d_Y(f(x_1), f(x_2)) \leq D \cdot r \cdot d_X(x_1, x_2).$$

The infimum of the numbers $D$ such that $f$ is a $D$-embedding is called the *distortion* of $f$. Every $n$-point metric space $(X, d)$ can be embedded in an $\mathcal{O}(\log n)$-dimensional Euclidean space with an $\mathcal{O}(\log n)$ distortion (Bourgain, 1985).

Johnson and Lindenstrauss obtained stronger control on the distortion at the cost of increasing the embedding dimension. Any set of $n$ points in a Euclidean space can be mapped to $\mathbb{R}^t$ where $t = \mathcal{O}(\frac{\log n}{\epsilon^2})$ with distortion at most $1 + \epsilon$ in the distances. Such a mapping may be found in random polynomial time (Johnson & Lindenstrauss, 1984).

Similar embedding theorems were obtained by Linial et al. in other $\ell_p$-spaces. In random polynomial-time $(X, d)$ may be embedded in $\ell_p^{\mathcal{O}(\log n)}$ (for any $1 \leq p \leq 2$), with distortion $\mathcal{O}(\log n)$ (Linial et al., 1995) or into $\ell_p^{\mathcal{O}(\log^2 n)}$ (for any $p > 2$), with distortion $\mathcal{O}(\log n)$ (Linial et al., 1995).

When the class of graphs is restricted, stronger embedding theorems are known. Krauthgamer et al. obtain embeddings with bounded distortion for graphs when certain minors are excluded; we mention the special case of planar graphs. Let $X$ be an $n$-point edge-weighted planar graph, equipped with the shortest path metric. Then $X$ embeds into $\ell_\infty^{\mathcal{O}(\log n)}$ with $\mathcal{O}(1)$ distortion (Krauthgamer et al., 2004).

Furthermore, there are results on the limitations of embedding graphs into $\ell_p$-spaces. For example, Linial et al. show that their embedding result for $1 \leq p \leq 2$ is sharp by considering expander graphs. Every embedding of an n-vertex constant-degree expander into an $\ell_p$ space, $2 \geq p \geq 1$, of any dimension, has distortion $\Omega(\log n)$. The metric space of such a graph cannot be embedded with constant distortion in any normed space of dimension $\mathcal{O}(\log n)$ (Linial et al., 1995).

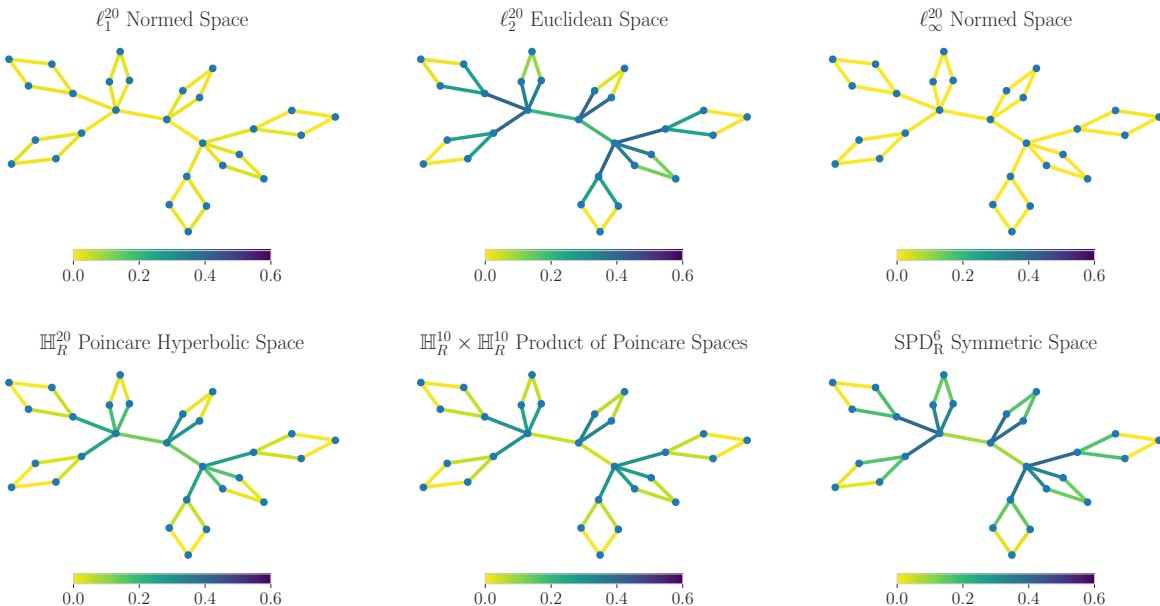

Figure 1: Embedding distortion across spaces on a small synthetic graph, with color indicating distortion levels (the absolute difference between graph edge and norm distances). The graph embeds well in the $\ell_1$ and $\ell_\infty$ normed spaces but endures distortion in other spaces.

These theoretical results illustrate the principle that large classes of finite metric spaces can in theory be abstractly embedded with low theoretical bounds on distortion in low dimensional normed spaces. Furthermore, the distortion and dimension can be substantially improved when the class of metric spaces is restricted. This leaves open many practical questions about the embeddability of real-world data into normed spaces and translating these theoretical results into predictions about the empirical results from experiments.

## 4 Experiments

We evaluate the graph embedding capacity of normed spaces alongside other popular Riemannian manifolds in the graph reconstruction task on various synthetic and real-world graphs (§4.1).

We analyze further (a) the space capacity and computational costs for varying graph sizes and curvatures; (b) space dimension; and in Appendix E, we extend our analysis to (c) expander graphs and (d) the asymmetry of the loss function. Additionally, we evaluate normed spaces in two tasks: link prediction (§4.2) and recommender systems (§4.3). For link prediction, we investigate the impact of normed spaces on **four popular graph neural networks**.

### 4.1 Benchmark: Graph Reconstruction

**Shortest Path Metric Embeddings.** A *metric embedding* is a mapping $f : X \to Y$ between two metric spaces $(X, d_X)$ and $(Y, d_Y)$. Ideally, one would desire metric embeddings to be distance preserving. In practice, accepting some distortion can be necessary. In this case, the overall quality of an embedding can be evaluated by *fidelity measures* such as the *average distortion $D_{avg}$* and the *mean average precision* mAP (cf. Appendix D.1 for the definitions). A special case of metric embedding is the *shortest path metric embedding*, also known as *low-distortion* or *approximate isometric* embedding, where $X$ is the node set $\mathcal{V}$ of a graph $\mathcal{G} = (\mathcal{V}, \mathcal{E})$ and $d_{\mathcal{G}}$ corresponds to the shortest path distance within $\mathcal{G}$. These embeddings represent or reconstruct the original graph $G$ in the chosen embedding space $Y$, ideally preserving the desirable geometric features of the graph.

**Learning Framework.** To compute these embeddings, we optimize a distance-based loss function inspired by generalized MDS (Bronstein et al., 2006) and which was used earlier in, e.g., Gu et al. (2019); López et al. (2021); Giovanni et al. (2022). Given graph distances $d_{\mathcal{G}}(u, v)$ between all pairs $u, v \in \mathcal{V}$ of nodes connected by a path in $\mathcal{G}$, which we denote $u \sim v$, the loss is defined as:

$$\mathcal{L}(f) = \sum_{u \sim v} \left| \left( \frac{d_Y(f(u), f(v))}{d_{\mathcal{G}}(u, v)} \right)^2 - 1 \right|, \tag{1}$$

where $d_Y(f(u), f(v))$ is the distance between the corresponding node embeddings in the target embedding space. In this context, the model parameters are a finite collection of points $f(u)$ in $Y$, each indexed by a specific node $u$ of $\mathcal{G}$. These parameters, i.e., the coordinates of the points, are optimized by minimizing the loss function through gradient descent. This loss function treats the distortion of different path lengths uniformly during training. We provide more context for the loss function in Appendix B. We also empirically evaluate embeddings learned by the distance-based loss function from eq. (1) against another popular distance-based loss function, namely mean squared error loss, in Appendix D.1.

**Motivation.** In geometric machine learning, graph reconstruction tasks have achieved a "de facto" benchmark status for empirically quantifying the representational capacity of geometric spaces for preserving graph structures given through their local close neighborhood information, global all-node interactions, or an intermediate of both (Nickel & Kiela, 2017; 2018; Gu et al., 2019; Cruceru et al., 2020; López et al., 2021; López et al., 2021). This fidelity to structure is crucial for downstream tasks such as link prediction and recommender systems, where knowing the relationships between nodes or users is key. Other applications include embedding large taxonomies. For instance, Nickel & Kiela (2017) and Nickel & Kiela (2018) proposed embedding WordNet while maintaining its local graph structure (semantic relationships between words), and applied these embeddings to downstream NLP tasks. Additionally, low-distortion metric embeddings have applications in approximation algorithms (Linial et al., 1995), online algorithms (Bansal et al., 2015), and distributed algorithms (Khan et al., 2008). We note that though we employ a global loss function, the resulting normed space embeddings preserve both local and global structure notably well.

**Experimental Setup.** Following the work of Gu et al. (2019), we train graph embeddings by minimizing the previously mentioned distance-based loss function. We follow López et al. (2021), and we do not apply any scaling to either the input graph distances or the distances calculated in the space, unlike earlier work (Gu et al., 2019; Cruceru et al., 2020). We report the average results across five runs in terms of (a) *average distortion $D_{avg}$* and (b) *mean average precision* (mAP). We provide the training details and the data statistics in Appendix D.1.

**Baseline Comparison.** We compare the performance of normed metric spaces with many other spaces for graph embedding. These spaces fall under three classes: (a) Normed spaces: $\mathbb{R}^{20}_{\ell_1}$, $\mathbb{R}^{20}_{\ell_2}$ and $\mathbb{R}^{20}_{\ell_\infty}$; (b) Riemannian symmetric spaces (Cruceru et al., 2020; López et al., 2021; López et al., 2021), incl. the space of SPD matrices: $\mathrm{SPD}^6_R$, Siegel upper half spaces: $\mathcal{S}^4_R$, $\mathcal{S}^4_{F_1}$, $\mathcal{S}^4_{F_\infty}$, bounded symmetric spaces: $\mathcal{B}^4_R$, $\mathcal{B}^4_{F_1}$, $\mathcal{B}^4_{F_\infty}$, hyperbolic spaces (Nickel & Kiela, 2017): $\mathbb{H}^{20}_R$ (Poincaré model), and product spaces (Gu et al., 2019): $\mathbb{H}^{10}_R \times \mathbb{H}^{10}_R$; (c) Cartesian product spaces involving normed spaces: $\mathbb{R}^{10}_{\ell_1} \times \mathbb{R}^{10}_{\ell_\infty}$, $\mathbb{R}^{10}_{\ell_1} \times \mathbb{H}^{10}_R$, $\mathbb{R}^{10}_{\ell_2} \times \mathbb{H}^{10}_R$ and $\mathbb{R}^{10}_{\ell_\infty} \times \mathbb{H}^{10}_R$; (d) pseudo-Euclidean space (Goldfarb, 1985; Vishwakarma & Sala, 2022): $\mathbb{R}^{10+,10-}_{\mathrm{PSE}}$. The notation for all metrics follows a standardized format: the superscript scales with the space dimension, and the subscript denotes the specific distance metric used (e.g., $R$ for Riemannian and $F$ for Finsler). Following López et al. (2021), we ensure uniformity across metric spaces by using the same number of free parameters, specifically a dimension of 20; and more importantly, we are concerned with the capacity of normed space at such low dimensions where non-Euclidean spaces have demonstrated success for embedding graphs (Chami et al., 2019b; Gu et al., 2019). We also investigate the capacities of spaces with growing dimensions, observing that other spaces necessitate much higher dimensions to match the capacity of the $\ell_\infty$ normed space (see Tab. 4). Note that $\mathcal{S}^n$ and $\mathcal{B}^n$ have $n(n+1)$ dimensions, and $\mathrm{SPD}^n$ has a dimension of $n(n+1)/2$. We elaborate on these metric spaces in Appendix A.

**Synthetic Graphs.** Following the work of López et al. (2021), we compare the representational capacity of various geometric spaces on several synthetic graphs, including grids, trees, and their Cartesian and rooted

| $(|V|, |E|)$ | 4D Grid (625, 2000) | | Tree (364, 363) | | Tree × Tree (225, 420) | | Tree ◇ Grid (775, 1270) | | Grid ◇ Tree (775, 790) | |
|---|---|---|---|---|---|---|---|---|---|---|
| | $D_{avg}$ | mAP | $D_{avg}$ | mAP | $D_{avg}$ | mAP | $D_{avg}$ | mAP | $D_{avg}$ | mAP |
| $\mathbb{R}^{20}_{\ell_1}$ | 1.08 ±0.00 | **100.00** | 1.62±0.02 | 73.56 | 1.22±0.01 | **100.00** | 1.22±0.01 | 71.91 | 1.75±0.02 | 60.13 |
| $\mathbb{R}^{20}_{\ell_2}$ | 11.24±0.00 | **100.00** | 3.92±0.04 | 42.30 | 9.78±0.00 | 96.03 | 3.86±0.02 | 34.21 | 4.28±0.04 | 27.50 |
| $\mathbb{R}^{20}_{\ell_\infty}$ | 0.13±0.00 | **100.00** | **0.15±0.01** | **100.00** | **0.58±0.01** | **100.00** | **0.09±0.01** | **100.00** | **0.23±0.02** | **99.39** |
| $\mathbb{H}^{20}_R$ | 25.23±0.05 | 63.74 | 0.54±0.02 | **100.00** | 20.59±0.11 | 75.67 | 14.56±0.27 | 44.14 | 14.62±0.13 | 30.28 |
| $\mathrm{SPD}^6_R$ | 11.24±0.00 | **100.00** | 1.79±0.02 | 55.92 | 8.83±0.01 | 98.49 | 1.56±0.02 | 62.31 | 1.83±0.00 | 72.17 |
| $\mathcal{S}^4_R$ | 11.27±0.01 | **100.00** | 1.35±0.02 | 78.53 | 8.68±0.02 | 98.03 | 1.45±0.09 | 72.49 | 1.54±0.08 | 76.66 |
| $\mathcal{S}^4_{F_\infty}$ | 5.92±0.06 | 99.61 | 1.23±0.28 | 99.56 | 3.31±0.06 | 99.95 | 10.88±0.19 | 63.52 | 10.48±0.21 | 72.53 |
| $\mathcal{S}^4_{F_1}$ | **0.01±0.00** | **100.00** | 0.76±0.02 | 91.57 | 1.08±0.16 | **100.00** | 1.03±0.00 | 78.71 | 0.84±0.06 | 80.52 |
| $\mathcal{B}^4_R$ | 11.28±0.01 | **100.00** | 1.27±0.05 | 74.77 | 8.74±0.09 | 98.12 | 2.88±0.32 | 72.55 | 2.76±0.11 | 96.29 |
| $\mathcal{B}^4_{F_\infty}$ | 7.32±0.16 | 97.92 | 1.51±0.13 | 99.73 | 4.26±0.26 | 99.70 | 6.55±1.77 | 73.80 | 7.15±0.85 | 90.51 |
| $\mathcal{B}^4_{F_1}$ | 0.39±0.02 | **100.00** | 0.77±0.02 | 94.64 | 1.28±0.16 | **100.00** | 1.09±0.03 | 76.55 | 0.99±0.01 | 81.82 |
| $\mathbb{R}^{10}_{\ell_1} \times \mathbb{R}^{10}_{\ell_\infty}$ | 0.16±0.00 | **100.00** | 0.63±0.02 | 99.73 | 0.62±0.00 | **100.00** | 0.54±0.01 | 99.84 | 0.60±0.01 | 94.81 |
| $\mathbb{R}^{10}_{\ell_1} \times \mathbb{H}^{10}_R$ | 0.55±0.00 | **100.00** | 1.13±0.01 | 99.73 | 0.62±0.01 | **100.00** | 1.76±0.02 | 50.74 | 1.65±0.01 | 89.47 |
| $\mathbb{R}^{10}_{\ell_2} \times \mathbb{H}^{10}_R$ | 11.24±0.00 | **100.00** | 1.19±0.04 | **100.00** | 9.30±0.04 | 98.03 | 2.15±0.05 | 58.23 | 2.03±0.01 | 97.88 |
| $\mathbb{R}^{10}_{\ell_\infty} \times \mathbb{H}^{10}_R$ | 0.14±0.00 | **100.00** | 0.22±0.02 | 96.96 | 1.91±0.01 | 99.13 | 0.15±0.01 | 99.96 | 0.57±0.01 | 90.34 |
| $\mathbb{H}^{10}_R \times \mathbb{H}^{10}_R$ | 18.74±0.01 | 78.47 | 0.65±0.02 | **100.00** | 8.61±0.03 | 97.63 | 1.08±0.06 | 77.20 | 2.80±0.65 | 84.88 |
| $\mathbb{R}^{10+,10-}_{\mathrm{PSE}}$ | 5.65±0.02 | 99.87 | 4.91±0.03 | 33.38 | 3.46±0.04 | 99.70 | 4.32±0.03 | 40.09 | 4.93±0.03 | 27.53 |

Table 1: Results on the five synthetic graphs. Lower $D_{avg}$ is better. Higher mAP is better. Metrics are given as percentages.

products. Further, we extend our analysis to three expander graphs, which can be considered theoretical worst-case scenarios for normed spaces embedding theorems (Linial et al., 1995), and thus are challenging setups for graph reconstruction. Tab. 1 and 2 report the results on synthetic graphs and expanders.

Overall, the $\ell_\infty$ normed space largely outperforms all other metric spaces considered on the graph configurations we examine. Notably, it excels over manifolds typically paired with specific graph topologies. For instance, the $\ell_\infty$ space significantly outperforms hyperbolic spaces, and surpasses Cartesian products of hyperbolic spaces and pseudo-Euclidean space on embedding tree graphs. Further, the $\ell_\infty$ space outperforms sophisticated symmetric spaces such as $\mathcal{S}^4_{F_1}$ and $\mathcal{B}^4_{F_1}$ on the graphs with mixed Euclidean and hyperbolic structures (Tree ◇ Grids and Grids ◇ Tree), although these symmetric spaces have compound geometries that combine Euclidean and hyperbolic subspaces. We also observe competitive performance from the $\ell_1$ space, which outperforms the $\ell_2$, hyperbolic, and symmetric spaces equipped with Riemannian and Finsler infinity metrics. Interestingly, combining $\ell_\infty$ and hyperbolic spaces using the Cartesian product does not bring added benefits and is less effective than using the $\ell_\infty$ space alone. Further, combining $\ell_1$ and $\ell_\infty$ spaces yields intermediate performance between the individual $\ell_1$ and $\ell_\infty$ spaces, due to the substantial performance gap between these two spaces. These findings underline the high capacity of the $\ell_1$ and $\ell_\infty$ spaces, aligning with our theoretical motivations.

In Tab. 2, we report graph reconstruction results for three expander graphs, namely Margulis-Gabber-Galil, Paley, and Chordal-Cycle graphs (Bollobás & Bollobás, 1998; Lubotzky, 1994; Vadhan et al., 2012). We note that expanders are considered representative of complex structures due to their high degree of sparsity and connectivity. As observed in our results, none of the metric spaces investigated align well with these intricate structures, leading to substantial distortion of graph structures across all spaces. Overall, graph structures in the $\ell_1$ and $\ell_\infty$ spaces endure much lower distortion in terms of $D_{avg}$ compared to the other spaces. Importantly, even with considerable distortion, the $\ell_1$ and $\ell_\infty$ spaces yield favorable mAP scores, particularly on Chordal. This suggests that the Chordal graph, while not isometrically embedded, is nearly isomorphic in these spaces, indicative of high-quality embeddings.

In sum, these results affirm that the $\ell_1$ and $\ell_\infty$ spaces are well-suited for embedding graphs, showing robust performance when their geometry closely, or even poorly, aligns with the graph structures.

**Real-World Graph Networks.** We evaluate the representational capacity of metric spaces on five popular real-world graph networks. These include (a) USCA312 (Hahsler & Hornik, 2007) and EuroRoad (Šubelj

| $(|V|,|E|)$ | MARGULIS (625, 2500) | | PALEY (101, 5050) | | CHORDAL (523, 1569) | |
|---|---|---|---|---|---|---|
| | $D_{avg}$ | mAP | $D_{avg}$ | mAP | $D_{avg}$ | mAP |
| $\mathbb{R}^{20}_{\ell_1}$ | **13.4±0.00** | **87.97** | 22.7±0.00 | 65.84 | 10.7±0.01 | **99.66** |
| $\mathbb{R}^{20}_{\ell_2}$ | 14.0±0.01 | 83.99 | 23.6±0.02 | 60.80 | 12.8±0.01 | 87.79 |
| $\mathbb{R}^{20}_{\ell_\infty}$ | 14.2±0.01 | 82.73 | **16.1±0.01** | **66.88** | 10.5±0.01 | 98.39 |
| $\mathbb{H}^{20}_R$ | 16.8±0.01 | 69.47 | 23.8±0.02 | 60.76 | 22.8±0.02 | 59.19 |
| $\text{SPD}^6_R$ | 14.1±0.01 | 84.98 | 23.6±0.01 | 61.76 | 12.8±0.01 | 77.59 |
| $\mathcal{S}^4_{F_1}$ | 24.2±0.02 | 2.24 | 26.6±0.01 | 51.94 | 38.1±0.02 | 1.40 |
| $\mathcal{B}^4_{F_1}$ | 24.1±0.01 | 2.17 | 26.5±0.01 | 52.97 | 37.2±0.01 | 1.43 |
| $\mathbb{R}^{10}_{\ell_1} \times \mathbb{R}^{10}_{\ell_\infty}$ | 13.8±0.00 | 87.25 | 20.4±0.01 | 60.09 | 10.6±0.01 | 99.47 |
| $\mathbb{R}^{10}_{\ell_1} \times \mathbb{H}^{10}_R$ | 14.2±0.00 | 83.63 | 23.3±0.00 | 62.77 | 11.7±0.00 | 82.95 |
| $\mathbb{R}^{10}_{\ell_2} \times \mathbb{H}^{10}_R$ | 14.4±0.00 | 79.12 | 23.7±0.01 | 60.72 | 12.8±0.00 | 81.93 |
| $\mathbb{R}^{10}_{\ell_\infty} \times \mathbb{H}^{10}_R$ | 14.6±0.01 | 86.26 | 20.8±0.00 | 60.57 | 12.1±0.01 | 88.98 |
| $\mathbb{H}^{10}_R \times \mathbb{H}^{10}_R$ | 15.4±0.01 | 75.77 | 23.7±0.01 | 60.33 | 17.2±0.00 | 58.25 |
| $\mathbb{R}^{10+,10-}_{\text{PSE}}$ | 15.7±0.02 | 45.25 | 24.7±0.01 | 56.78 | **9.1±0.02** | 76.48 |

Table 2: Results on the three expander graphs. Metrics are given as percentages.

| $(|V|,|E|)$ | USCA312 (312, 48516) | BIO-DISEASOME (516, 1188) | | CSPHD (1025, 1043) | | EUROROAD (1039, 1305) | | FACEBOOK (4039, 88234) | |
|---|---|---|---|---|---|---|---|---|---|
| | $D_{avg}$ | $D_{avg}$ | mAP | $D_{avg}$ | mAP | $D_{avg}$ | mAP | $D_{avg}$ | mAP |
| $\mathbb{R}^{20}_{\ell_1}$ | 0.29±0.01 | 1.62±0.01 | 89.14 | 1.59±0.02 | 52.34 | 1.73±0.01 | 93.61 | 2.38±0.02 | 31.22 |
| $\mathbb{R}^{20}_{\ell_2}$ | **0.18±0.01** | 3.83±0.01 | 76.31 | 4.04±0.01 | 47.37 | 4.50±0.00 | 87.70 | 3.16±0.01 | 32.21 |
| $\mathbb{R}^{20}_{\ell_\infty}$ | 0.95±0.02 | **0.53±0.01** | **98.24** | **0.42±0.01** | **99.28** | **1.06±0.01** | **99.48** | **0.71±0.02** | 42.21 |
| $\mathbb{H}^{20}_R$ | 2.39±0.02 | 6.83±0.08 | 91.26 | 22.42±0.23 | 60.24 | 43.56±0.44 | 54.25 | 3.72±0.00 | 44.85 |
| $\text{SPD}^6_R$ | 0.21±0.02 | 2.54±0.00 | 82.66 | 2.92±0.11 | 57.88 | 19.54±0.99 | 92.38 | 2.92±0.05 | 33.73 |
| $\mathcal{S}^4_R$ | 0.28±0.03 | 2.40±0.02 | 87.01 | 4.30±0.18 | 59.95 | 29.21±0.91 | 84.92 | 3.07±0.04 | 30.98 |
| $\mathcal{S}^4_{F_\infty}$ | 0.57±0.08 | 2.78±0.49 | 93.95 | 27.27±1.00 | 59.45 | 46.82±1.02 | 72.03 | 1.90±0.11 | **45.58** |
| $\mathcal{S}^4_{F_1}$ | **0.18±0.02** | 1.55±0.04 | 90.42 | 1.50±0.03 | 64.11 | 3.79±0.07 | 94.63 | 2.37±0.07 | 35.23 |
| $\mathcal{B}^4_R$ | 0.24±0.07 | 2.69±0.10 | 89.11 | 28.65±3.39 | 62.66 | 53.45±2.65 | 48.75 | 3.58±0.10 | 30.35 |
| $\mathcal{B}^4_{F_\infty}$ | 0.21±0.04 | 4.58±0.63 | 90.36 | 26.32±6.16 | 54.94 | 52.69±2.28 | 48.75 | 2.18±0.18 | 39.15 |
| $\mathcal{B}^4_{F_1}$ | **0.18±0.07** | 1.54±0.02 | 90.41 | 2.96±0.91 | 67.58 | 21.98±0.62 | 91.63 | 5.05±0.03 | 39.87 |
| $\mathbb{R}^{10}_{\ell_1} \times \mathbb{R}^{10}_{\ell_\infty}$ | 0.47±0.01 | 1.56±0.01 | 98.22 | 1.38±0.02 | 89.18 | 1.65±0.02 | 98.34 | 2.16±0.02 | 39.90 |
| $\mathbb{R}^{10}_{\ell_1} \times \mathbb{H}^{10}_R$ | 0.72±0.01 | 1.99±0.01 | 93.78 | 1.83±0.02 | 78.10 | 2.26±0.02 | 96.19 | 2.77±0.02 | 33.79 |
| $\mathbb{R}^{10}_{\ell_2} \times \mathbb{H}^{10}_R$ | **0.18±0.00** | 2.52±0.02 | 91.99 | 3.06±0.02 | 73.25 | 4.24±0.02 | 89.93 | 2.80±0.01 | 34.26 |
| $\mathbb{R}^{10}_{\ell_\infty} \times \mathbb{H}^{10}_R$ | 0.42±0.02 | 1.42±0.02 | 96.51 | 1.16±0.01 | 76.91 | 1.77±0.01 | 97.38 | 1.41±0.02 | 35.03 |
| $\mathbb{H}^{10}_R \times \mathbb{H}^{10}_R$ | 0.47±0.18 | 2.57±0.05 | 95.00 | 7.02±1.07 | 79.22 | 23.30±1.62 | 75.07 | 2.51±0.00 | 36.39 |
| $\mathbb{R}^{10+,10-}_{\text{PSE}}$ | 0.41±0.01 | 3.87±0.02 | 70.82 | 3.17±0.02 | 35.41 | 2.49±0.01 | 90.33 | 5.10±0.03 | 20.36 |

Table 3: Results on the five real-world graphs. Metrics are given as percentages.

& Bajec, 2011), representing North American city networks and European road systems respectively; (b) BIO-DISEASOME (Goh et al., 2007), a biological graph representing the relationships between human disorder and diseases and their genetic origins; (c) CSPHD (Nooy et al., 2011), a graph of Ph.D. advisor-advisee relationships in computer science and (d) FACEBOOK (McAuley & Leskovec, 2012), a dense social network from Facebook.

In Tab. 3, the $\ell_1$ and $\ell_\infty$ spaces generally outperform all other metric spaces on real-world graphs, consistent with the synthetic graph results. However, for USCA312—a weighted graph of North American cities where edge lengths match actual spherical distances—the inherent spherical geometry limits effective embedding into the $\ell_1$ and $\ell_\infty$ spaces at lower dimensions.

**Graph Representational Capacity.** We assess the capacity of the metric spaces for embedding graphs of increasing size, focusing on trees (negative curvature), grids (zero curvature), and fullerenes (positive curvature). See the illustrations of these graphs in Appendix F. For trees, we fix the valency at 3 and vary the tree height from 1 to 7; for 4D grids, we vary the grid dimension from 2 to 7; for fullerenes, we vary the number of carbon atoms from 20 to 240. We report the average results across three runs.

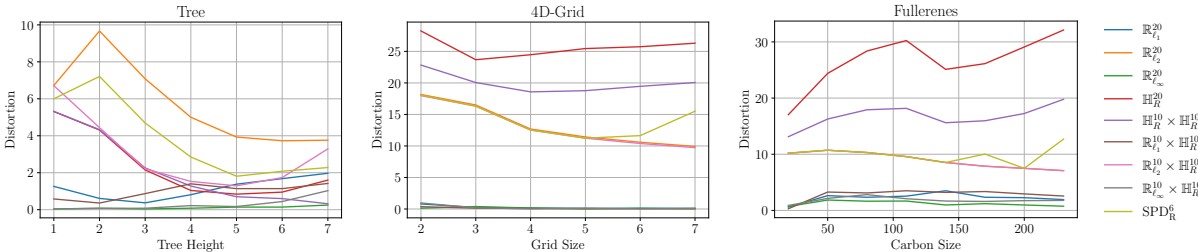

Figure 2: Metric space capacities with growing graph size and unnoticeable distortion variance.

Fig. 2 (left) reports the results on trees. Within the range of computationally feasible graphs, we see that as graph size grows, the capacity of all metric spaces, barring the $\ell_\infty$ space and its product with hyperbolic space, improves significantly before plateauing. In hyperbolic spaces, which are not scale-invariant, embedding trees with high fidelity to all path lengths could require scaling (see, e.g., Sarkar (2012), Sala et al. (2018)). This can be seen in the high embedding distortion of small trees, specifically those with a height less than 4. Further empirical analysis demonstrates that the optimization goal localizes unavoidable distortion to some paths of short combinatorial lengths and their contribution to the average loss becomes smaller with increased size since there are relatively fewer of them. In contrast, the $\ell_\infty$ space consistently exhibits a high capacity, largely unaffected by graph size, and significantly outperforms the hyperbolic space within the observed range.

Fig. 2 (center) reports the results on 4D grids with zero curvature. We find that the metric spaces whose geometry aligns poorly with grid structures, such as the $\ell_2$ space, the hyperbolic space and their products, exhibit weak representational capacity. In contrast, the $\ell_1$ and $\ell_\infty$ spaces preserve grid structures consistently well as the graph size increases. Fig. 2 (right) reports the results on fullerenes with positive curvature. Given that none of the spaces considered feature a positively curved geometry, they are generally ill-suited for embedding fullerenes. However, we see that the $\ell_1$ and $\ell_\infty$ spaces and the product spaces accommodating either of these two spaces consistently outperform others even as the number of carbon atoms increases.

Overall, these results show that the $\ell_1$ and $\ell_\infty$ spaces consistently surpass other metric spaces in terms of representation capacity. They exhibit small performance fluctuation across various curvatures and maintain robust performance within graph configurations and size ranges we consider.

**Training Efficiency.** Fig. 3 compares the training time for different metric spaces on grids and trees with growing size. The training time grows as $C(\text{space}) \times \mathcal{O}(\text{number of paths})$, where $C(\text{space})$ is a constant that depends on the embedding space. Among these spaces, SPD demands the highest amount of training efforts, even when dealing with small grids and trees. For other spaces, the training time differences become more noticeable with increasing graph size. The largest difference appears at a grid size of 7 and a tree height of 6: The $\ell_1$, $\ell_2$, and $\ell_\infty$ normed spaces exhibit the highest efficiency, outperforming product, hyperbolic and SPD spaces in training time.

These results are expected given that transcendental functions and eigendecompositions are computationally costly operations. Overall, normed spaces show high scalability with increasing graph size. Their training time grows much slower than product spaces and Riemannian alternatives.

**Space Dimension.** Tab. 4 compares the results of different spaces across dimensions on the BIO-DISEASOME dataset. The surveyed theoretical results suggest that in sufficiently high dimensions, space capacities appear to approach theoretical limits, leading to the possibility that the performance of different spaces can become similar. However, we find that other spaces necessitate very high dimensions to match the capacity of the $\ell_\infty$ normed space. For instance, even after tripling space dimension, $\mathbb{H}_R^{66}$ and $\text{SPD}_R^{11}$ still perform much worse than $\mathbb{R}_{\ell_\infty}^{20}$. $\mathbb{R}_{\ell_1}^{66}$ rivals $\mathbb{R}_{\ell_\infty}^{20}$ only in mAP. $\mathbb{R}_{\ell_1}^{33} \times \mathbb{R}_{\ell_\infty}^{33}$ surpasses $\mathbb{R}_{\ell_\infty}^{20}$ in mAP but lags behind in $D_{avg}$. López et al. (2021) similarly evaluated Euclidean, hyperbolic and spherical spaces, their products, and Siegel space at $n = 306$. Their best results on BIO-DISEASOME were $D_{avg} = 0.73$ and mAP = 99.09 for $\mathcal{S}_{F_1}^{17}$. In contrast,

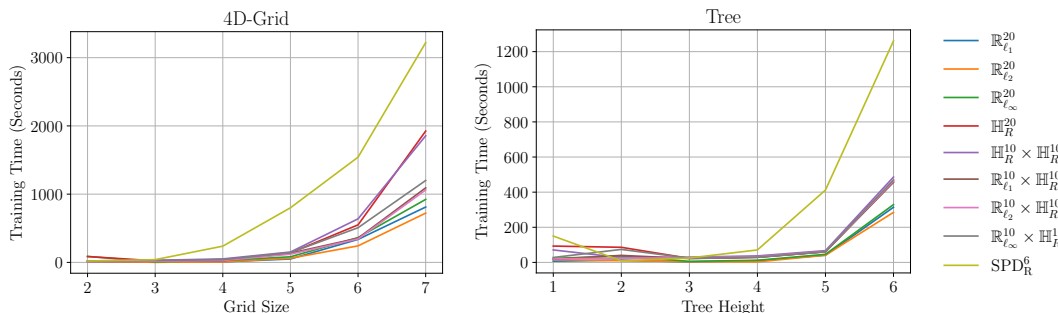

Figure 3: Training time scales up as the graph size increases.

$\ell_\infty^{20}$ and $\ell_1^{18} \times \ell_\infty^{18}$ achieved $D_{\text{avg}} = 0.5 \pm 0.01$ and mAP $= 99.4$, respectively. These results show that normed spaces efficiently yield low-distortion embeddings at much lower dimensions than other spaces.

| | $n = 20$ | | $n = 36$ | | $n = 66$ | |
|---|---|---|---|---|---|---|
| | $D_{avg}$ | mAP | $D_{avg}$ | mAP | $D_{avg}$ | mAP |
| $\mathbb{R}^n_{\ell_1}$ | 1.6±0.01 | 89.1 | 1.6±0.01 | 94.3 | 1.7±0.01 | 98.1 |
| $\mathbb{R}^n_{\ell_2}$ | 3.8±0.01 | 76.3 | 3.8±0.01 | 85.9 | 3.9±0.01 | 86.2 |
| $\mathbb{R}^n_{\ell_\infty}$ | **0.5±0.01** | **98.2** | **0.5±0.01** | 98.3 | **0.6±0.01** | 99.2 |
| $\mathbb{H}^n_R$ | 6.8±0.08 | 91.2 | 5.8±0.06 | 93.6 | 5.9±0.05 | 93.2 |
| $\text{SPD}^k_R$ | 2.5±0.00 | 82.6 | 2.4±0.02 | 87.8 | 2.3±0.02 | 90.5 |
| $\mathbb{R}^{\frac{n}{2}}_{\ell_1} \times \mathbb{R}^{\frac{n}{2}}_{\ell_\infty}$ | 1.5±0.01 | **98.2** | 1.2±0.01 | **99.4** | 1.4±0.01 | **99.8** |
| $\mathbb{R}^{\frac{n}{2}}_{\ell_1} \times \mathbb{H}^{\frac{n}{2}}_R$ | 1.9±0.01 | 93.7 | 1.8±0.01 | 95.8 | 1.7±0.01 | 98.4 |
| $\mathbb{R}^{\frac{n}{2}}_{\ell_2} \times \mathbb{H}^{\frac{n}{2}}_R$ | 2.5±0.02 | 91.9 | 2.6±0.02 | 92.3 | 2.5±0.01 | 94.6 |
| $\mathbb{R}^{\frac{n}{2}}_{\ell_\infty} \times \mathbb{H}^{\frac{n}{2}}_R$ | 1.4±0.02 | 96.5 | 1.1±0.02 | 98.6 | 0.8±0.01 | 98.8 |
| $\mathbb{H}^{\frac{n}{2}}_R \times \mathbb{H}^{\frac{n}{2}}_R$ | 2.5±0.05 | 95.0 | 2.5±0.04 | 97.4 | 2.6±0.05 | 97.6 |

Table 4: Results on BIO-DISEASOME. When $n$ takes 20, 36, 66, $k$ in $\text{SPD}^k_R$ takes 6, 8, 11. Metrics are given as percentages.

## 4.2 Application 1: Link Prediction

**Experimental Setup.** We comparably evaluate the impact of normed spaces on four popular architectures of graph neural networks (GNNs), namely GCN (Kipf & Welling, 2017), GAT (Veličković et al., 2018), SGC (Wu et al., 2019) and GIN (Xu et al., 2019). Following Kipf & Welling (2016); Chami et al. (2019a), we evaluate GNNs in the link prediction task on two citation network datasets: Cora and Citeseer (Sen et al., 2008). This task aims to predict the presence of edges (links) between nodes that are not seen during training. We split each dataset into train, development and test sets corresponding to 70%, 10%, 20% of citation links that we sample at random. We report the average performance in AUC across five runs. We provide the training details in Appendix D.2.

**Results.** Tab. 5 reports the results of GNNs across different spaces for link prediction. While previous works showed the superiority of hyperbolic over Euclidean space for GNNs at lower dimensions on these datasets (Chami et al., 2019a; Zhao et al., 2023), our findings indicate the opposite (see the results from $\mathbb{H}^n_R$ and $\mathbb{R}^n_{\ell_2}$). This is attributed to the vanishing impact of hyperbolic space when operating GNNs in a larger dimensional space (with up to 128 dimension to achieve optimal performance on development sets). The $\ell_1$ normed space consistently outperforms other spaces (including the $\ell_\infty$ and product spaces), demonstrating its superiority for link prediction.

| | CORA | | | | CITESEER | | | |
|---|---|---|---|---|---|---|---|---|
| | GCN | GAT | SGC | GIN | GCN | GAT | SGC | GIN |
| $\mathbb{R}^n_{\ell_1}$ | **93.4±0.3** | **92.8±0.4** | **93.7±0.5** | **91.6±0.5** | **93.1±0.3** | **93.1±0.4** | 93.8±0.4 | **92.4±0.3** |
| $\mathbb{R}^n_{\ell_2}$ | 92.1±0.5 | 91.7±0.5 | 91.1±0.3 | 90.2±0.5 | 91.4±0.5 | 91.1±0.4 | 93.8±0.4 | 92.0±0.3 |
| $\mathbb{R}^n_{\ell_\infty}$ | 89.5±0.4 | 88.2±0.5 | 88.8±0.3 | 88.4±0.5 | 90.3±0.4 | 89.5±0.5 | 91.7±0.3 | 90.5±0.3 |
| $\mathbb{H}^n_R$ | 86.1±0.5 | 92.1±0.6 | 89.9±0.4 | 87.7±0.3 | 92.5±0.2 | 91.1±0.3 | 91.0±0.3 | 91.7±0.4 |
| $\mathbb{R}^{\frac{n}{2}}_{\ell_1} \times \mathbb{R}^{\frac{n}{2}}_{\ell_\infty}$ | 93.0±0.5 | 92.3±0.3 | 93.5±0.5 | 90.9±0.4 | 92.9±0.5 | 92.8±0.6 | **94.6±0.5** | **92.4±0.4** |
| $\mathbb{R}^{\frac{n}{2}}_{\ell_1} \times \mathbb{H}^{\frac{n}{2}}_R$ | 89.5±0.5 | 90.7±0.4 | 88.7±0.4 | 89.0±0.6 | 91.5±0.4 | 90.3±0.3 | 90.6±0.4 | 90.3±0.5 |
| $\mathbb{R}^{\frac{n}{2}}_{\ell_2} \times \mathbb{H}^{\frac{n}{2}}_R$ | 90.2±0.4 | 90.6±0.5 | 88.3±0.6 | 87.8±0.4 | 90.8±0.5 | 91.2±0.3 | 91.0±0.5 | 90.4±0.3 |
| $\mathbb{R}^{\frac{n}{2}}_{\ell_\infty} \times \mathbb{H}^{\frac{n}{2}}_R$ | 89.7±0.5 | 90.7±0.3 | 89.2±0.3 | 87.7±0.4 | 90.5±0.3 | 90.2±0.4 | 90.8±0.4 | 90.4±0.3 |
| $\mathbb{H}^{\frac{n}{2}}_R \times \mathbb{H}^{\frac{n}{2}}_R$ | 85.2±0.3 | 88.0±0.4 | 88.7±0.4 | 87.6±0.3 | 90.4±0.3 | 87.2±0.5 | 89.2±0.3 | 91.3±0.5 |

Table 5: Results of GNNs in different spaces for link prediction, where $n$ is a hyperparameter of space dimension that we tune on the development sets. Results are reported in AUC.

| | ML-100K | | LASTFM | | MEETUP | |
|---|---|---|---|---|---|---|
| | HR@10 | nDG | HR@10 | nDG | HR@10 | nDG |
| $\mathbb{R}^{20}_{\ell_1}$ | 54.5±1.2 | 28.2 | **69.3±0.4** | 48.9 | **82.1±0.4** | **63.3** |
| $\mathbb{R}^{20}_{\ell_2}$ | 54.6±1.0 | 28.7 | 55.4±0.3 | 24.6 | 79.8±0.2 | 59.5 |
| $\mathbb{R}^{20}_{\ell_\infty}$ | 50.1±1.1 | 25.5 | 54.9±0.5 | 31.7 | 70.2±0.2 | 45.3 |
| $\mathbb{H}^{20}_R$ | 53.4±1.0 | 28.2 | 54.8±0.5 | 24.9 | 79.1±0.5 | 58.8 |
| $\mathrm{SPD}^6_R$ | 53.3±1.4 | 28.0 | 55.4±0.2 | 25.3 | 78.5±0.5 | 58.6 |
| $\mathcal{S}^4_{F_1}$ | **55.6±1.3** | **29.4** | 61.1±1.2 | 38.0 | 80.4±0.5 | 61.1 |
| $\mathbb{R}^{10}_{\ell_1} \times \mathbb{R}^{10}_{\ell_\infty}$ | 52.0±1.1 | 27.1 | 68.2±0.4 | 47.3 | 79.6±0.3 | 60.1 |
| $\mathbb{R}^{10}_{\ell_1} \times \mathbb{H}^{10}_R$ | 53.1±1.2 | 27.6 | 69.2±0.5 | **49.9** | 80.6±0.3 | 61.2 |
| $\mathbb{R}^{10}_{\ell_2} \times \mathbb{H}^{10}_R$ | 53.1±1.3 | 27.9 | 45.5±0.4 | 18.9 | 79.3±0.2 | 58.9 |
| $\mathbb{R}^{10}_{\ell_\infty} \times \mathbb{H}^{10}_R$ | 54.9±1.2 | 28.4 | 66.2±0.5 | 48.2 | 77.8±0.4 | 57.2 |
| $\mathbb{H}^{10}_R \times \mathbb{H}^{10}_R$ | 54.8±0.9 | 29.1 | 55.0±0.6 | 24.6 | 79.5±0.2 | 59.2 |

Table 6: Results on the three recommendation bipartite graphs. Higher HR@10 and nDG are better.

### 4.3 Application 2: Recommender Systems

**Experimental Setup.** Following López et al. (2021), we conduct a comparative examination of the impact of the choice of metric spaces on a recommendation task. This task can be seen as a binary classification problem on a bipartite graph, in which users and items are treated as two distinct subsets of nodes, and recommendation systems are tasked with predicting the interactions between user-item pairs. We adopt the approach of prior research (Vinh Tran et al., 2020; López et al., 2021) and base recommendation systems on graph embeddings in metric spaces. Our experiments include three popular datasets: (a) ML-100K (Harper & Konstan, 2015) from MovieLens for movie recommendation; (b) LAST.FM (Cantador et al., 2011) for music recommendation, and (c) MEETUP (Pham et al., 2015) from Meetup.com in NYC for event recommendation. We use the train/dev/test sets of these datasets from the work of López et al. (2021), and report the average results across five runs in terms of two evaluation metrics: hit ratio (HR) and normalized discounted cumulative gain (nDG), both at 10. We provide the training details and the statistics of the graphs in Appendix D.3.

**Results.** Tab. 6 reports the results on three bipartite graphs. We find that the performance gaps between metric spaces are small on ML-100K. Therefore, the choice of metric spaces does not influence much the performance on this graph. In contrast, the gaps are quite noticeable on the other two graphs. For instance, we see that the $\ell_1$ space largely outperforms all the other spaces, particularly on LAST.FM. This showcases the importance of choosing a suitable metric space for downstream tasks. It is noteworthy that the $\ell_1$ norm outperforms the $\ell_\infty$ norm on the recommender systems task, while the opposite is true for the graph reconstruction task. This raises intriguing questions about how normed space embeddings leverage the geometries of the underlying normed spaces.

## 5 Conclusions

Classical discrete geometry results suggest that normed spaces can abstractly embed a wide range of finite metric spaces, including graphs, with surprisingly low theoretical bounds on distortion. Motivated by these theoretical insights, we highlight normed spaces as a valuable complement to popular manifolds for graph representation learning. Our empirical findings show that normed spaces consistently outperform other manifolds across several real-world and synthetic graph reconstruction benchmark datasets. Notably, normed spaces demonstrate an enhanced capacity to embed graphs of varying curvatures, an increasingly evident advantage as graph sizes get bigger. We further illustrate the practical utility of normed spaces on two applied graph embedding tasks, namely link prediction and recommender systems, underscoring their potential for applications. Moreover, while delivering superior performance, normed spaces require significantly fewer computational resources and pose fewer technical challenges than competing solutions, further enhancing their appeal. Our work not only emphasizes the importance of normed spaces for graph representation learning but also naturally raises several questions and motivates further research directions:

**Modern and Classical AI/ML Applications.** The potential of normed space embeddings can be tested across a wide range of AI applications. In many machine learning applications, normed spaces provide a promising alternative to existing Riemannian manifolds, such as hyperbolic spaces (Nickel & Kiela, 2017; 2018; Chami et al., 2020a;b) and other symmetric spaces, as embedding spaces. Classical non-differentiable discrete methods for embedding graphs into normed spaces have found applications in various areas (Livingston & Stout, 1988; Linial et al., 1995; Deza & Shtogrin, 2000; Mohammed, 2005). Our work demonstrates the efficient computation of graph embeddings into normed spaces using a modern differentiable programming paradigm. Integrating normed spaces into deep learning frameworks holds the potential to advance graph representation learning and its applications, bridging modern and classical AI research.

**Discrete Geometry.** Further analysis is needed to describe how normed space embeddings leverage the geometry of normed spaces. It is also important to investigate which emergent geometric properties of the embeddings can be used for analyzing graph structures, such as hierarchies. Lastly, we anticipate our work will provide a valuable experimental mathematics tool.

**Limitations and Future Research.** Our work and others in the geometric machine literature, such as Nickel & Kiela (2017; 2018); Chami et al. (2019a); Cruceru et al. (2020); López et al. (2021); Giovanni et al. (2022), lack theoretical guarantees. It is crucial to connect the theoretical bounds on distortion for abstract embeddings and the empirical results, especially for real-world graphs. In particular, it is possible that hyperparameter tuning or different spaces and training techniques than normed spaces and gradient descent can achieve better performance in general settings, in addition to the restricted cases where some geometric spaces perform exceptionally well on particular graphs, such as hyperbolic geometry and trees. It would also be valuable to analyze more growing families of graphs, such as expanders and mixed-curvature graphs. Furthermore, embedding larger real-world networks would provide insights into scalability in practical settings, and how the benefits would translate to graphs of larger scale. Lastly, future work should expand this study to dynamic graphs with evolving structures and investigating the transferability of embeddings learned on one task, e.g., link predictions with GNNs, to other tasks.

## Acknowledgments

We would like to extend our thanks to Anna Wienhard, Maria Beatrice Pozetti, Ullrich Koethe, Federico López, and Steve Trettel for many interesting conversations and valuable insights. We extend our sincere gratitude to the reviewers for their insightful comments and suggestions, which have significantly enhanced the quality of this paper. J.M. Riestenberg was supported by the RTG 2229 "Asymptotic Invariants and Limits of Groups and Spaces" and by the DFG under Project-ID 338644254 - SPP2026. Wei Zhao was supported by the Klaus Tschira Foundation and a Young Marsilius Fellowship at Heidelberg University.

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

# A Embedding Spaces

**Metric Spaces.** Let $X$ be a non-empty set. A *metric space* is an ordered pair $(X, d)$, where $d : X \times X \to \mathbb{R}$ is a function, called the *metric* or *distance function*, that satisfies the following properties for all $x, y, z \in X$: (i) $d(x, y) \geq 0$, (ii) $d(x, y) = 0$ if and only if $x = y$, (iii) $d(x, y) = d(y, x)$, and (iv) $d(x, z) \leq d(x, y) + d(y, z)$. A map $f : X \to Y$ between two metric spaces $(X, d_X)$ and $(Y, d_Y)$ is an *isometric embedding* if it preserves distances, i.e., $d_Y(f(x_1), f(x_2)) = d_X(x_1, x_2), \forall\, x_1, x_2 \in X$.

**Riemannian Manifolds.** Let $M$ be a smooth manifold, $p \in M$ be a point, and $T_pM$ be the tangent space at the point $p$. A Riemannian manifold $(M, g)$ is a smooth manifold $M$ equipped with a Riemannian metric $g$ given by a smooth inner product $g_p : T_pM \times T_pM \to \mathbb{R}$ at each point $p \in M$. Euclidean space is the simplest example of a Riemannian manifold. Let $V$ be any $n$-dimensional real vector space endowed with the Euclidean metric $g$ given by $g(v, w) = \langle v, w \rangle$ for any $p \in V$ and any $v, w \in T_pV \cong V$.

**Normed Spaces.** A *normed space* is a vector space $V$ over the real numbers $\mathbb{R}$ or complex numbers $\mathbb{C}$ equipped with a norm. A *norm* is a function $\|\cdot\| : V \to [0, +\infty)$ satisfying the following properties for all vectors $x, y \in V$ and scalars $\alpha \in \mathbb{F}$: (i) $\|x\| \geq 0$, with equality if and only if $x = 0$, (ii) $\|\alpha x\| = |\alpha| \|x\|$, and (iii) $\|x + y\| \leq \|x\| + \|y\|$. Normed spaces induce metric spaces via the *induced distance function*, defined as $d(x, y) = \|x - y\|$. The $p$-norms are among the most important examples of norms. For a real number $p \geq 1$, the $p$-norm of a vector $x \in \mathbb{R}^d$ is given by $\|x\|_p := (|x_1|^p + |x_2|^p + \cdots + |x_d|^p)^{\frac{1}{p}}$. The definition is extended for $p = \infty$ as $\|x\|_\infty := \max_{1 \leq i \leq d} |x_i|$. The space $\mathbb{R}^d$ equipped with $p$-norm is denoted as $\ell_p^d$. Here we focus on the cases $p = 1, 2$, and $\infty$.

**Pseudo-Euclidean Spaces.** Denote the product $\mathbb{R}^{d^+} \times \mathbb{R}^{d^-}$, where $d^+$ and $d^-$ are non-negative integers, by $\mathbb{R}_{\mathrm{PSE}}^{d^+, d^-}$, and write any element $x \in \mathbb{R}_{\mathrm{PSE}}^{d^+, d^-}$ as $x = (x^+, x^-)$, where $x^+ \in \mathbb{R}^{d^+}$ and $x^- \in \mathbb{R}^{d^-}$. Then the *pseudo-Euclidean space* $\mathbb{R}_{\mathrm{PSE}}^{d^+, d^-}$ is the set $\mathbb{R}^{d^+} \times \mathbb{R}^{d^-}$ with the squared distance function defined by $d_{\mathrm{PSE}}^2(x, y) = \|x^+ - y^+\|_2^2 - \|x^- - y^-\|_2^2$, where $\|\cdot\|_2^2$ is the square of the $\ell_2$ norm on the respective space.

**Hyperbolic Space.** Hyperbolic space is a Riemannian manifold with a constant negative curvature. There are several models of hyperbolic space, such as the Poincaré ball model and Lorentz model. The models are essentially the same in a mathematical sense (they are pairwise isometric), but one model can have computational advantages over another.

**Definition 1** (Poincaré Ball Model)**.** *Let* $\|\cdot\|$ *be the Euclidean norm. Given a negative curvature c, the Poincaré ball model is a Riemannian manifold* $(\mathcal{B}_c^n, g_{\mathbf{x}}^{\mathcal{B}})$*, where* $\mathcal{B}_c^n = \{\mathbf{x} \in \mathbb{R}^n : \|\mathbf{x}\|^2 < -1/c\}$ *is an open ball with radius* $1/\sqrt{|c|}$ *and* $g_{\mathbf{x}}^{\mathcal{B}} = (\lambda_{\mathbf{x}}^c)^2 \mathrm{Id}$*, where* $\lambda_{\mathbf{x}}^c = 2/(1 + c\|\mathbf{x}\|_2^2)$ *and* $\mathrm{Id}$ *is the identity matrix.*

**Product Manifold.** Let $M_1, M, \ldots, M_k$ be a sequence of smooth manifolds. The product manifold is given by the Cartesian product $M = M_1 \times M_2 \times \cdots \times M_k$. Each point $p \in M$ has the coordinates $p = (p_1, \ldots, p_k)$, with $p_i \in M_i$ for all $i$. Similarly, a tangent vector $v \in T_pM$ can be written as $(v_1, \ldots, v_k)$, with each $v_i \in T_{p_i}M_i$. If each $M_i$ is equipped with a Riemannian metric $g_i$, then the product manifold $M$ can be given the product metric where $g(v, w) = \sum_{i=1}^k g_i(v_i, w_i)$.

**Riemannian Symmetric Spaces.** Riemannian symmetric spaces are connected Riemannian manifolds such that the geodesic symmetry[2] at each point defines a global isometry of the space. For simply connected manifolds this condition is equivalent to having covariantly constant curvature tensor. A key consequence of the definition is that symmetric spaces are homogeneous manifolds. Intuitively, this means that the manifold "looks the same" at every point. Furthermore, simply connected symmetric spaces decompose into products of irreducible symmetric spaces and Euclidean space. Irreducible symmetric spaces can be described in terms of semisimple Lie groups. Basic examples of Riemannian symmetric spaces include Euclidean spaces, hyperbolic

---

[2]For any point $p$ in any Riemannian manifold, there exists a sufficently small $\epsilon > 0$ such that the map $S_p : B(p, \epsilon) \to B(p, \epsilon)$ defined by $S_p(c(t)) = c(-t)$ is well-defined for any unit-speed geodesic $c : (-\epsilon, \epsilon) \to B(p, \epsilon)$ with $c(0) = p$. Such a map $S_p$ is called the *geodesic symmetry* at $p$.

| | Space | Underlying Set | Distance |
|---|---|---|---|
| $\mathbb{R}^n_{\ell_p}$ | normed space | $\mathbb{R}^n$ | $d(x,y) = \|x-y\|_p$ |
| $\mathbb{H}^n_R$ | hyperbolic space | $\{\mathbb{R}^n \mid \|x\|_2 \le 1\}$ | $d(x,y) = \operatorname{arcosh}\left(1 + 2\frac{\|x-y\|_2^2}{(1-\|x\|_2^2)(1-\|y\|_2^2)}\right)$ |
| $\text{SPD}^k_R$ | SPD (López et al., 2021) | symmetric positive definite $k \times k$ matrices | $d(x,y) = \|(\lambda_i(x^{-1}y))_{i=1}^k\|_2$, where $\lambda_i(x^{-1}y)$ is the $i$th eigenvalue of $x^{-1}y$ |
| *Siegel spaces* (López et al., 2021) | | | |
| $\mathcal{S}^k_R$ | upper half model | $\{Z = X + iY \in \operatorname{Sym}(n,\mathbb{C}) \mid Y >> 0\}$ | see Algorithm 1 |
| $\mathcal{S}^k_{F_\infty}$ | upper half model | $\{Z = X + iY \in \operatorname{Sym}(n,\mathbb{C}) \mid Y >> 0\}$ | see Algorithm 1 |
| $\mathcal{S}^k_{F_1}$ | upper half model | $\{Z = X + iY \in \operatorname{Sym}(n,\mathbb{C}) \mid Y >> 0\}$ | see Algorithm 1 |
| $\mathcal{B}^k_R$ | bounded symmetric domain model | $\{Z \in \operatorname{Sym}(n,\mathbb{C}) \mid Id - Z^*Z >> 0\}$ | see Algorithm 1 |
| $\mathcal{B}^k_R$ | bounded symmetric domain model | $\{Z \in \operatorname{Sym}(n,\mathbb{C}) \mid Id - Z^*Z >> 0\}$ | see Algorithm 1 |
| $\mathcal{B}^k_R$ | bounded symmetric domain model | $\{Z \in \operatorname{Sym}(n,\mathbb{C}) \mid Id - Z^*Z >> 0\}$ | see Algorithm 1 |
| $\mathbb{R}^{d^+,d^-}_{\text{PSE}}$ | pseudo-Euclidean space (Vishwakarma & Sala, 2022) | $\mathbb{R}^{d^+} \times \mathbb{R}^{d^-}$ | $d(x,y) = \sqrt{\|x^+ - y^+\|_2^2 - \|x^- - y^-\|_2^2}$ (when defined), where $x = (x^+, x^-)$ and $y = (y^+, y^-)$ |
| $\mathcal{M}_1 \times \mathcal{M}_2$ | product space | $\mathcal{M}_1 \times \mathcal{M}_2$ | $d(x,y) = \sqrt{d_1(x_1,y_1)^2 + d_2(x_2,y_2)^2}$, where $x = (x_1, x_2)$ and $y = (y_1, y_2)$ |

Table 7: A summary of the embedding spaces.

spaces and spheres. In the following we will describe two further special cases: Siegel space and the space of symmetric positive definite (SPD) matrices.

Siegel spaces, $\text{HypSPD}_n$, are matrix versions of the hyperbolic plane, accommodating many products of hyperbolic planes and the copies of SPD as submanifolds. These spaces support Finsler metrics that induce the $\ell_1$ and the $\ell_\infty$ metric on the Euclidean subspaces. $\text{HypSPD}_n$ has the two following models with $n(n+1)$ dimensions, both of which are open subsets of the space $\operatorname{Sym}(n,\mathbb{C})$ over $\mathbb{C}$. These two models generalize the Poincaré disk and the upper half plane model of the hyperbolic space.

**Definition 2** (Bounded Symmetric Domain Model). *The bounded symmetric domain model for $HypSPD_n$ generalizes the Poincaré disk. It is given by $\mathcal{B}_n := \{Z \in \operatorname{Sym}(n,\mathbb{C}) \mid \operatorname{Id} - Z^*Z \gg 0\}$.*

**Definition 3** (Siegel Upper Half Space Model). *The Siegel upper half space model for $HypSPD_n$ generalizes the upper half plane model of the hyperbolic plane by $\mathcal{S}_n := \{Z = X + iY \in \operatorname{Sym}(n,\mathbb{C}) \mid Y \gg 0\}$.*

There exists an isomorphism from $\mathcal{B}_n$ to $\mathcal{S}_n$ given by the Cayley transform, which is a matrix analogue of the familiar map from the Poincare disk to upper half space model of the hyperbolic plane:

$$Z \mapsto i(Z + \operatorname{Id})(Z - \operatorname{Id})^{-1}.$$

We refer readers to Siegel (1943) and López et al. (2021) for an in-depth overview of Siegel spaces and their applications in graph embeddings.

**Definition 4** (SPD Space). $\text{SPD}_n$ *is the space of positive definite real symmetric $n \times n$ matrices, given by $\operatorname{SPD}(n,\mathbb{R}) := \{X \in \operatorname{Sym}(n,\mathbb{R}) \mid X \gg 0\}$. It has the structure of a Riemannian manifold of non-positive curvature of $n(n+1)/2$ dimensions. The Riemannian metric on $\operatorname{SPD}_n$ is defined as follows: if $U, V \in S_n$ are tangent vectors based at $P \in \operatorname{SPD}_n$, their inner product is given by $\langle U, V \rangle_P = \operatorname{Tr}(P^{-1}UP^{-1}V)$.*

The tangent space to any point of $\text{SPD}_n$ can be identified with the vector space $S_n$ of all real symmetric $n \times n$ matrices. $\text{SPD}_n$ is more flexible than Euclidean or hyperbolic geometries, or products thereof. In particular, it contains $n$-dimensional Euclidean subspaces, $(n-1)$-dimensional hyperbolic subspaces, products of $\lfloor \frac{n}{2} \rfloor$ hyperbolic planes, and many other interesting spaces as totally geodesic submanifolds, see the reference (Helgason, 1978) for an in-depth introduction.

## B Graph Reconstruction Loss Function

The graph reconstruction task aims to empirically quantify the capacity of a space for embedding graph structure given through its node-to-node shortest paths. Recent work has generally employed local, global, or hybrid loss functions, focusing on close neighborhood information, all-node interactions, or an intermediate of

| Finite Metric Spaces | Embedding Space | Distortion Bound | Reference |
|---|---|---|---|
| Complete graph $(K_n)$ | $l_1^{\lceil \log_2(n) \rceil}$ | $\mathcal{O}(1)$ | (Linial et al., 1995) |
| Tree $(T_n)$ | $\ell_\infty^{\mathcal{O}(\log n)}$ | $\mathcal{O}(1)$ | (Linial et al., 1995) |
| Planar graph with $n$ vertices | $\ell_\infty^{\mathcal{O}(\log n)}$ | $\mathcal{O}(1)$ | (Krauthgamer et al., 2004) |
| Expander with $n$ vertices | $\ell_p$ of any dimension $(2 \geq p \geq 1)$ | $\Omega(\log n)$ | (Linial et al., 1995) |
| Metric space $(X, d)$ with $n$ points | $\ell_p^{\mathcal{O}(\log n)}$ (for any $1 \leq p \leq 2$) | $\mathcal{O}(\log n)$ | (Linial et al., 1995) |
| Metric space $(X, d)$ with $n$ points | $\ell_p^{\mathcal{O}(\log^2 n)}$ (for any $p > 2$) | $\mathcal{O}(\log n)$ | (Linial et al., 1995) |

Table 8: A summary of theoretical results.

both. Local loss functions emphasize preserving neighborhoods, exemplified by the loss function

$$\mathcal{L}(f) = - \sum_{(u,v) \in E} \log \frac{\exp\left(-d_Y(f(u), f(v))\right)}{\sum_{w \in \mathcal{N}(u)} \exp\left(-d_Y(f(u), f(w))\right)}.$$

from Nickel & Kiela (2017; 2018), where $\mathcal{N}(u) = \{w \mid (u, w) \notin E\} \cup \{v\}$ is the set of negative examples for $u$ (including $v$). The resulting embeddings are typically favored by rank-based evaluation metrics such as mean average precision mAP. On the other hand, global functions emphasize preserving distances directly via loss functions motivated by generalized MDS (Bronstein et al. (2006)), exemplified by the loss function

$$\mathcal{L}(f) = \sum_{u \sim v} \left| \left( \frac{d_Y(f(u), f(v))}{d_{\mathcal{G}}(u, v)} \right)^2 - 1 \right|,$$

from Gu et al. (2019), and which we use in this work. The resulting embeddings are typically favored by average distortion $D_{avg}$. Lastly, hybrid loss functions, such as the Riemannian Stochastic Neighbor Embedding (RSNE) from Cruceru et al. (2020), aim to balance the emphasis on local and global, sometimes with a tunable parameter for controlling the optimization goal.

We note that though we employ a global loss function, the resulting normed space embeddings notably perform well on both $D_{avg}$ and mAP.

## C   Metric Learning

Metric learning is a machine learning approach concerned with learning distance metrics in an embedding space, with the aim of using the distances between data points as features for tasks such as classification, regression, clustering, and image retrieval. In connection with our work, metric learning pipelines broadly consist of a learnable embedding function $f : X \to Y$ that maps data from an input space $X$ into a target embedding space $Y$ equipped with a distance metric $d_Y$, a machine learning component that takes the values of the distance $d_f(\cdot, \cdot) := d_Y(f(\cdot), f(\cdot))$ as input features for a task, and appropriate optimization algorithms for learning the parameters in the pipeline. The embedding function and the parameters of the machine learning components (if any) could be jointly learned in a supervised manner or otherwise hand-crafted or leveraged from embeddings trained on another task. In *deep* metric learning, typically, the embedding function and the machine learning component are differentiable, and parameter optimization takes place by minimizing or maximizing an appropriate loss function with gradient descent. For reference, we recommend the following excellent surveys on metric learning: Kulis et al. (2013); Kaya & Bilge (2019). We note that many geometric machine learning pipelines, including ours, align with the metric learning paradigm, where the metric space has a Riemannian manifold structure. Check, for example, Nickel & Kiela (2017; 2018); Chami et al. (2019a); Cruceru et al. (2020); López et al. (2021); Giovanni et al. (2022).

We summarize the metric learning pipelines used in this work in Table 9. A *shallow embedding* is a function that simply assigns each entity to a point in the target embedding space, with these points serving as the learnable parameters. On the other hand, a *graph neural network* maps the features of a node and its neighbors to a point in the target space, with the network weights being the learnable parameters.

| Task | Embedding Function | ML Component | Loss Function |
|---|---|---|---|
| **Graph Reconstruction** (§4.1, D.1) | shallow embedding | — | distance-based loss (eq. 1) |
| **Link Prediction** (§4.2, D.2) | graph neural network | Fermi-Dirac decoder (eq. 4) | binary cross-entropy (eq. 5) |
| **Recommender System** (§4.3, D.3) | shallow embedding | similarity score (eq. 6) | hinge loss (eq. 7) binary cross-entropy (eq. 8) |

Table 9: Summary of metric learning pipelines.

| Graph | Nodes | Edges | Triples | Grid Layout | Tree Valency | Tree Height |
|---|---|---|---|---|---|---|
| 4D GRID | 625 | 2000 | 195,000 | $(5)^4$ | - | - |
| TREE | 364 | 363 | 66,066 | - | 3 | 5 |
| TREE × TREE | 225 | 420 | 25,200 | - | 2 | 3 |
| TREE ◇ GRIDS | 775 | 1,270 | 299,925 | 5 × 5 | 2 | 4 |
| GRID ◇ TREES | 775 | 790 | 299,925 | 5 × 5 | 2 | 4 |

Table 10: Characteristics of synthetic graphs.

# D  Experiments

**Hardware and Code Release.** All experiments were executed on an Intel(R) Xeon(R) CPU E5-2650 computer, equipped with 48 CPUs operating at 2.2 GHz and a single Tesla P40 GPU with a 24GB of memory running on CUDA 11.2.

## D.1  Graph Reconstruction

**Implementation Details.** In all setups, we use the RADAM optimizer (Bécigneul & Ganea, 2019), and run the same grid search to to train graph embeddings. The implementation of all baselines are taken from Geoopt (Kochurov et al., 2020) and López et al. (2021). We train for 3000 epochs, and stop training when the average distortion has not decreased for 200 epochs. We experiment with three hyperparameters: (a) learning rate $\in \{0.1, 0.01, 0.001\}$; (b) batch size $\in \{512, 1024, 2048, -1\}$ with $-1$ as the node count within a graph and (c) maximum gradient norm $\in \{10, 50, 250\}$. Table 10 and 11 report the stats of all the synthetic and real-world graphs.

**Evaluation Metrics.** We evaluate the quality of the learned embeddings using distortion and precision metrics. Consider a graph $G$, a target metric space $Y$, and a metric embedding $f : G \to Y$. The distortion of the embedding of a pair of nodes $u, v$ is given by:

$$\text{distortion}(u, v) = \frac{|d_Y(f(u), f(v)) - d_G(u, v)|}{d_G(u, v)}.$$

| Graph | Nodes | Edges | Triples |
|---|---|---|---|
| USCA312 | 312 | 48,516 | 48,516 |
| BIO-DISEASOME | 516 | 1,188 | 132,870 |
| CSPHD | 1,025 | 1,043 | 524,800 |
| ROAD-EUROROAD | 1,039 | 1,305 | 539,241 |
| FACEBOOK | 4,039 | 88,234 | 8,154,741 |
| MARGULIS | 625 | 2,500 | 195,000 |
| PALEY | 101 | 5,050 | 5,050 |
| CHORDAL | 523 | 1,569 | 136,503 |

Table 11: Characteristics of real-world and expander graphs.

| $(|V|, |E|)$ | BIO-DISEASOME (516, 1188) | | CSPHD (1025, 1043) | |
|---|---|---|---|---|
| | $D_{avg}$ | mAP | $D_{avg}$ | mAP |
| *Stress Loss* | | | | |
| $\mathbb{R}^{20}_{\ell_1}$ | 2.79±0.01 | 87.09 | 2.16±0.01 | 45.55 |
| $\mathbb{R}^{20}_{\ell_2}$ | 4.41±0.02 | 76.71 | 4.51±0.01 | 39.05 |
| $\mathbb{R}^{20}_{\ell_\infty}$ | **1.88±0.01** | **88.86** | **1.54±0.01** | **69.00** |
| $\mathbb{H}^{20}_R$ | 11.34±0.05 | 66.55 | 30.88±0.06 | 19.62 |
| *Distortion Loss* | | | | |
| $\mathbb{R}^{20}_{\ell_1}$ | 1.62±0.01 | 89.14 | 1.59±0.02 | 52.34 |
| $\mathbb{R}^{20}_{\ell_2}$ | 3.83±0.01 | 76.31 | 4.04±0.01 | 47.37 |
| $\mathbb{R}^{20}_{\ell_\infty}$ | **0.53±0.01** | **98.24** | **0.42±0.01** | **99.28** |
| $\mathbb{H}^{20}_R$ | 6.83±0.08 | 91.26 | 22.42±0.23 | 60.24 |

Table 12: Comparison of distortion and stress loss functions.

We denote the average of distortion over all pairs of nodes by $D_{avg}$.

The other metric that we consider is the mean average precision (mAP). It is a ranking-based measure for local neighborhoods that does not track explicit distances. For the mean average precision (mAP) metric, consider $G = (V, E)$ as a graph and $\mathcal{N}_a$ as the neighborhood of the node $a \in V$. Let $R_{a,b_i}$ be the smallest neighborhood of $f(a)$ in the space $Y$ that contains $f(b_i)$, with $f: G \to \mathcal{P}$ as a metric embedding. Then, mAP can be defined as follows:

$$\text{mAP}(f) = \frac{1}{|V|} \sum_{a \in V} \frac{1}{\deg(a)} \sum_{i=1}^{|\mathcal{N}_a|} \frac{|\mathcal{N}_a \cap R_{a,b_i}|}{|R_{a,b_i}|}.$$

mAP quantifies how well the embedding approximates graph isomorphism, applicable only to unweighted graphs. mAP measures the average discrepancy between the neighborhood of each node $u \in V$ and the neighborhood of $f(u) \in Y$. It's important to note that an embedding with zero average distortion guarantees a perfect mean average precision score (i.e., 100.00), but the inverse is not always true: an embedding that effectively preserves the adjacency structure might not be an isometry.

**Distortion Loss vs. Stress Loss.** We note that distance-based loss functions inspired by generalized MDS (Bronstein et al., 2006) include distortion and stress loss functions. *Distortion loss*, which is the main loss function we use for learning embeddings in this work, is given by:

$$\mathcal{L}_{\text{distortion}}(f) = \sum_{u \sim v} \left| \left( \frac{d_Y(f(u), f(v))}{d_{\mathcal{G}}(u, v)} \right)^2 - 1 \right|. \tag{2}$$

(Compare eq. (2) with eq. (1).) On the other hand, *strain loss*, also known as *mean squared error loss*, is given by:

$$\mathcal{L}_{\text{stress}}(f) = \sum_{u \sim v} \left( d_{\mathcal{G}}(u, v) - d_Y(f(u), f(v)) \right)^2. \tag{3}$$

In Table 12, we compare the impacts of distortion loss and mean squared error loss functions on graph embeddings for graph reconstruction. Our results show that with respect to the $D_{avg}$ and mAP evaluation metrics, $\ell_\infty$ embeddings trained with both loss functions consistently outperform the other embeddings, and for each space, the embeddings trained with the distortion loss outperform the embeddings trained with mean squared error. This justifies our choice for learning embeddings with distortion loss.

**Space Dimension.** In Table 14, 15, 16, and 13, we vary the dimension size on datasets with varying curvatures. Our results show that for $\ell_1$ and $\ell_\infty$, performance generally improves as dimension grows bigger.

We also observe that increasing the dimension does not seem to help much when there is mismatch between the geometry of a space and a graph, such as is the case with $\mathbb{R}^n_{\ell_2}$ and $\mathbb{H}^n_{\ell_R}$ on Grid, and $\mathbb{H}^n_{\ell_R}$ on Fullerenes. Lastly, we highlight that López et al. (2021) similarly evaluated embeddings on the BIO-DISEASOME dataset in Euclidean, hyperbolic and spherical spaces, their products, and Siegel space at dimension $n = 306$; our low-dimensional normed spaced embeddings outperformed their high-dimensional embeddings.

**Choice of Norm.** The choice of the norm could be considered a hyperparameter to be tuned. In our experiments, $\ell_\infty$ performs best in the graph reconstruction task on 10/13 datasets, whereas for downstream tasks (link prediction and recommender systems), $\ell_1$ performs consistently best in almost all cases on five datasets. So, we recommend using $\ell_\infty$ for graph reconstruction and $\ell_1$ for downstream tasks. Alternatively, we recommend the product of $\ell_1$ and $\ell_\infty$ in all setups, as that product performs consistently as the second best option behind $\ell_1$ and $\ell_\infty$.

**Link Prediction.** We evaluate the performance of task-agnostic shortest-path metric embeddings in a link prediction task. Here, *task-agnostic* and *task-specific* refer to the technique used for training the embeddings. We first embed the nodes of the CORA and CITESEER datasets by minimizing the distance-based distortion loss defined in eq. (1) using a training set of existing edges. We note that no node features are used in this process. Subsequently, we train a logistic regression classifier on the Hadamard product of source and target node embeddings to predict whether a link exists between the nodes for a training set that includes existing and non-existing edges. Lastly, we evaluate the classifier on a test set of existing and non-existing edges. We follow the same experimental setup for link prediction from Appendix D.2 and tune hyperparameters (including dimension size and learning rate) on the development set.

Table 17 compares the results for link prediction using the task-agnostic and task-specific embeddings on the CORA and CITESEER datasets. (The results for the task-specific embeddings are taken from Table 5.) We observe that $\ell_1$ space performs best among task-agnostic embeddings on these link prediction datasets, but overall, task-agnostic embeddings underperform task-specific counterparts in our setup. Even though shortest-path metric embeddings capture enough information to enable non-trivial accuracy in link prediction, they fall short of capturing the more nuanced information present in node features and higher-order proximity, resulting in performance that lags behind task-specific embeddings that do capture the aforementioned information. Thus, among the embeddings considered for link prediction in this work, we recommend task-specific $\ell_1$ GNN embeddings for practitioners.

| | $n = 20$ | | $n = 36$ | | $n = 66$ | |
|---|---|---|---|---|---|---|
| | $D_{avg}$ | mAP | $D_{avg}$ | mAP | $D_{avg}$ | mAP |
| $\mathbb{R}^n_{\ell_1}$ | 1.59±0.02 | 52.34 | 1.32±0.01 | 68.35 | 1.11±0.01 | 82.40 |
| $\mathbb{R}^n_{\ell_2}$ | 4.04±0.01 | 47.37 | 3.84±0.01 | 62.35 | 3.77±0.02 | 68.86 |
| $\mathbb{R}^n_{\ell_\infty}$ | 0.42±0.01 | 99.28 | 0.50±0.01 | 99.16 | 0.47±0.01 | 99.57 |
| $\mathbb{H}^n_R$ | 22.42±0.23 | 60.24 | 21.81±0.20 | 74.62 | 21.54±0.15 | 75.45 |

Table 13: Results on CSPHD.

| | $n = 20$ | | $n = 36$ | | $n = 66$ | |
|---|---|---|---|---|---|---|
| | $D_{avg}$ | mAP | $D_{avg}$ | mAP | $D_{avg}$ | mAP |
| $\mathbb{R}^n_{\ell_1}$ | 1.08±0.00 | 100.00 | 0.36±0.00 | 100.00 | 0.21±0.00 | 100.00 |
| $\mathbb{R}^n_{\ell_2}$ | 11.24±0.00 | 100.00 | 11.23±0.00 | 100.00 | 11.22±0.00 | 100.00 |
| $\mathbb{R}^n_{\ell_\infty}$ | 0.13±0.00 | 100.00 | 0.02±0.00 | 100.00 | 0.02±0.00 | 100.00 |
| $\mathbb{H}^n_R$ | 25.23±0.05 | 63.74 | 25.30±0.05 | 68.69 | 25.25±0.05 | 68.78 |

Table 14: Results on GRID (zero curvature).

| | $n = 20$ | | $n = 36$ | | $n = 66$ | |
|---|---|---|---|---|---|---|
| | $D_{avg}$ | mAP | $D_{avg}$ | mAP | $D_{avg}$ | mAP |
| $\mathbb{R}^n_{\ell_1}$ | $1.62\pm0.02$ | 73.56 | $0.96\pm0.02$ | 90.61 | $0.68\pm0.02$ | 97.80 |
| $\mathbb{R}^n_{\ell_2}$ | $3.92\pm0.04$ | 42.30 | $3.13\pm0.02$ | 55.19 | $2.77\pm0.02$ | 55.59 |
| $\mathbb{R}^n_{\ell_\infty}$ | $0.15\pm0.01$ | 100.00 | $0.02\pm0.01$ | 100.00 | $0.02\pm0.01$ | 100.00 |
| $\mathbb{H}^n_R$ | $0.54\pm0.02$ | 100.00 | $0.43\pm0.02$ | 100.00 | $0.51\pm0.02$ | 100.00 |

Table 15: Results on TREE (negative curvature).

| | $n = 20$ | | $n = 36$ | | $n = 66$ | |
|---|---|---|---|---|---|---|
| | $D_{avg}$ | mAP | $D_{avg}$ | mAP | $D_{avg}$ | mAP |
| $\mathbb{R}^n_{\ell_1}$ | $3.32\pm0.02$ | 100.00 | $3.06\pm0.03$ | 100.00 | $2.97\pm0.03$ | 100.00 |
| $\mathbb{R}^n_{\ell_2}$ | $8.53\pm0.03$ | 100.00 | $8.45\pm0.03$ | 100.00 | $8.25\pm0.03$ | 100.00 |
| $\mathbb{R}^n_{\ell_\infty}$ | $2.95\pm0.02$ | 100.00 | $1.96\pm0.01$ | 100.00 | $1.59\pm0.01$ | 100.00 |
| $\mathbb{H}^n_R$ | $25.18\pm0.05$ | 84.13 | $25.34\pm0.04$ | 84.51 | $25.22\pm0.05$ | 84.89 |

Table 16: Results on FULLERENES-140 (positive curvature).

## D.2 Link Prediction

**Implementation Details.** For each dataset, we use grid search to tune hyperparameters on the development set. Our hyperparameters include (a) dimension $\in \{32, 64, 128\}$ and (b) learning rate $\in \{0.1, 0.01, 0.001\}$. We set batch size to the number of nodes present in each graph dataset. We train for 1000 epochs and stop training when the loss on the development set has not been decreased for 200 epochs. We report the average performance in AUC across five runs. Following Chami et al. (2019a), we use the Fermi-Dirac decoder to compute the likelihood of a link between node pairs, and generate negative sets by randomly selecting links from non-connected node pairs. All graph neural networks are trained by optimizing the cross-entropy loss function. We extend the implementation of Poincaré GCN (Chami et al., 2019a) to support the other three architectures, enabling them to operate in both hyperbolic space and product spaces. We reduce the learning rate by a factor of 5 if GNNs cannot improve the performance after 50 epochs for hyperbolic and product spaces.

**Model.** Given a graph $\mathcal{G} = (\mathcal{V}, \mathcal{E})$ with a vertex set $\mathcal{V}$ and edge set $\mathcal{E}$, node features $\mathbf{x}_u \in \mathbb{R}^d$ for each node $u \in \mathcal{V}$, and a target metric space $(Z, d_Z)$, a GNN is used to map each node $u$ to an embedding $\mathbf{z}_u \in Z$. The Fermi-Dirac decoder is used to compute probability scores for edges:

$$p_{u,v} = \left(1 + \exp\left(\frac{d_Z^2(\mathbf{z}_u, \mathbf{z}_v) - r}{t}\right)\right)^{-1}, \tag{4}$$

where $p_{u,v}$ is the probability of an edge existing between nodes $u$ and $v$, $d_Z(\mathbf{z}_u, \mathbf{z}_v)$ is the distance between the embeddings of the nodes, $r$ is a learnable parameter that adjusts the decision boundary, and $t$ is a learnable temperature parameter that controls the sharpness of the decision boundary.

**Loss Function.** Given a training set $\mathcal{E}_{\text{train}} := \mathcal{E}_{\text{pos}} \cup \mathcal{E}_{\text{neg}}$ consists of existing edges $\mathcal{E}_{\text{pos}}$ and non-existing edges $\mathcal{E}_{\text{neg}}$, the binary cross-entropy loss used to train the model is given by:

$$\mathcal{L} = -\left(\sum_{(u,v)\in\mathcal{E}_{\text{pos}}} \log(\sigma(p_{u,v})) + \sum_{(u,v)\in\mathcal{E}_{\text{neg}}} \log(1 - \sigma(p_{u,v}))\right), \tag{5}$$

where $p_{u,v}$ is the output of the model and $\sigma(x) = \frac{1}{1+e^{-x}}$ is the sigmoid function.

**Space Dimension.** In Table 18, we vary the dimension size on the Cora dataset for link prediction. Our results show that, $\ell_1$ performs consistently best across GNNs and dimensions from 32 to 128, and overall, the results show improvement with increased dimension.

| Embedding | CORA | CITESEER |
|---|---|---|
| *Low-Distortion Embeddings* (task-agnostic) | | |
| $\mathbb{R}^n_{\ell_1}$ | 81.3±0.3 | 76.3±0.5 |
| $\mathbb{R}^n_{\ell_2}$ | 79.5±0.4 | 73.8±0.3 |
| $\mathbb{R}^n_{\ell_\infty}$ | 77.0±0.3 | 76.3±0.5 |
| *GCN Embeddings* (task-specific) | | |
| GCN in $\mathbb{R}^n_{\ell_1}$ | 93.4±0.3 | 93.1±0.3 |
| GCN in $\mathbb{R}^n_{\ell_2}$ | 92.1±0.5 | 91.4±0.5 |
| GCN in $\mathbb{R}^n_{\ell_\infty}$ | 89.5±0.4 | 90.3±0.4 |

Table 17: Results for link prediction on CORA and CITESEER using different embeddings.

| | $n=32$ | | | | $n=64$ | | | | $n=128$ | | | |
|---|---|---|---|---|---|---|---|---|---|---|---|---|
| | GCN | GAT | SGC | GIN | GCN | GAT | SGC | GIN | GCN | GAT | SGC | GIN |
| $\mathbb{R}^n_{\ell_1}$ | **93.4±0.3** | 91.2±0.3 | 92.5±0.3 | 90.2±0.4 | 92.1±0.3 | 92.2±0.3 | **93.7±0.5** | **91.6±0.5** | 92.5±0.3 | **92.8±0.4** | 93.0±0.3 | 91.0±0.3 |
| $\mathbb{R}^n_{\ell_2}$ | 91.3±0.3 | 91.1±0.2 | 90.7±0.1 | **90.2±0.5** | 91.5±0.5 | 90.9±0.5 | 90.8±0.3 | 89.5±0.5 | **92.1±0.5** | 91.7±0.5 | **91.1±0.3** | 89.2±0.5 |
| $\mathbb{R}^n_{\ell_\infty}$ | 89.0±0.4 | 86.2±0.2 | 87.6±0.3 | 87.4±0.4 | 89.1±0.4 | 87.2±0.5 | **88.8±0.3** | **88.4±0.5** | 89.5±0.4 | 88.2±0.5 | 87.8±0.3 | 87.2±0.5 |
| $\mathbb{H}^n_R$ | 84.8±0.3 | 84.3±0.2 | 87.7±0.3 | 86.6±0.3 | 86.1±0.3 | 89.0±0.5 | 88.7±0.3 | 85.9±0.3 | 86.1±0.5 | **92.1±0.6** | **89.9±0.4** | **87.7±0.3** |

Table 18: Results on CORA for link prediction.

## D.3 Recommender Systems

**Implementation Details.** We follow a metric learning approach Vinh Tran et al. (2020), with the implementation of all baselines taken from López et al. (2021). We minimize the hinge loss function for ML-100K and LAST.FM, while minimizing the binary cross-entropy (BCE) function for MEETUP. We use the RSGD optimizer (Bonnabel, 2011) to tune graph node embeddings. In all setups, we run the same grid search to train recommender systems. We train for 500 epochs, reduce the learning rate by a factor of 5 if the model does not improve the performance after 50 epochs. We stop training when the loss on the dev set has not been decreased for 50 epochs. We use the burn-in strategy (Nickel & Kiela, 2017; Cruceru et al., 2020) that trains recommender systems with a 10 times smaller learning rate for the first 10 epochs. We experiment with three hyperparameters: (a) learning rate $\in \{0.1, 0.01, 0.001\}$; (b) batch size $\in \{512, 1024, 2048\}$ and (c) maximum gradient norm $\in \{5, 10, 50\}$. Table 19 reports the stats of all the bipartite graphs in the recommendation task.

**Model.** Given a set of entities $\mathcal{E}$ and a target metric space $(X, d_X)$, we associate with each entity $e \in \mathcal{E}$ an embedding $f(e) \in X$ and bias terms $b_{e,\text{lhs}}, b_{e,\text{rhs}} \in \mathbb{R}$, where $f : \mathcal{E} \to X$ is a learnable embedding function. Given a pair of entities $e_1, e_2 \in \mathcal{E}$, the model computes a similarity score $\phi(e_1, e_2)$ as follows

$$\phi_{f,b,X}(e_1, e_2) := b_{e_1,\text{lhs}} + b_{e_2,\text{rhs}} - d_X^2(f(e_1), f(e_2)). \tag{6}$$

Subtracting the square distance ensures that the entities whose embeddings are closer in the metric space have a higher score, making it a suitable representation of similarity. The model we use is *shallow*: It learns a collection of points $f(e) \in M$ indexed by the entities $e \in \mathcal{E}$. In our setting, $\mathcal{E} = \mathcal{U} \cup \mathcal{V}$, where $\mathcal{U}$ is the space of users and $\mathcal{V}$ is the space of items.

**Hinge Loss Function.** Given a set $\mathcal{T} = \{(u, v)\}$ of observed user-item interactions, the hinge loss function is given by:

$$\mathcal{L} = \sum_{(u,v) \in \mathcal{T}} \sum_{(u,w) \notin \mathcal{T}} [m + \phi_{f,b,X}(u, v) - \phi_{f,b,X}(u, w)]_+, \tag{7}$$

where $w$ is an item the user $u$ has not interacted with, $m$ is the hinge margin, and $[z]_+ = max(0, z)$. For each user $u$, we generate a negative set by randomly selecting 100 items that the user has not interacted with.

**Binary Cross-Entropy Loss Function.** Let $\mathcal{T}_1$ and $\mathcal{T}_2$ be a set of observed user-item interactions and a set of non-interactions, respectively. Consider $\mathcal{T} = \mathcal{T}_1 \cup \mathcal{T}_2$ as the collection of all interactions and non-interactions. For each pair $(u, v) \in \mathcal{T}$, let $y_{u,v} \in \{0, 1\}$ denote the true label: If the pair belongs to $\mathcal{T}_1$, then $y_{u,v} = 1$,

| Dataset | Users | Items | Interactions | Density (%) |
|---|---|---|---|---|
| ML-100K | 943 | 1,682 | 100,000 | 6.30 |
| LAST.FM | 1,892 | 17,632 | 92,834 | 0.28 |
| MEETUP-NYC | 46,895 | 16,612 | 277,863 | 0.04 |

Table 19: Recommender system dataset stats

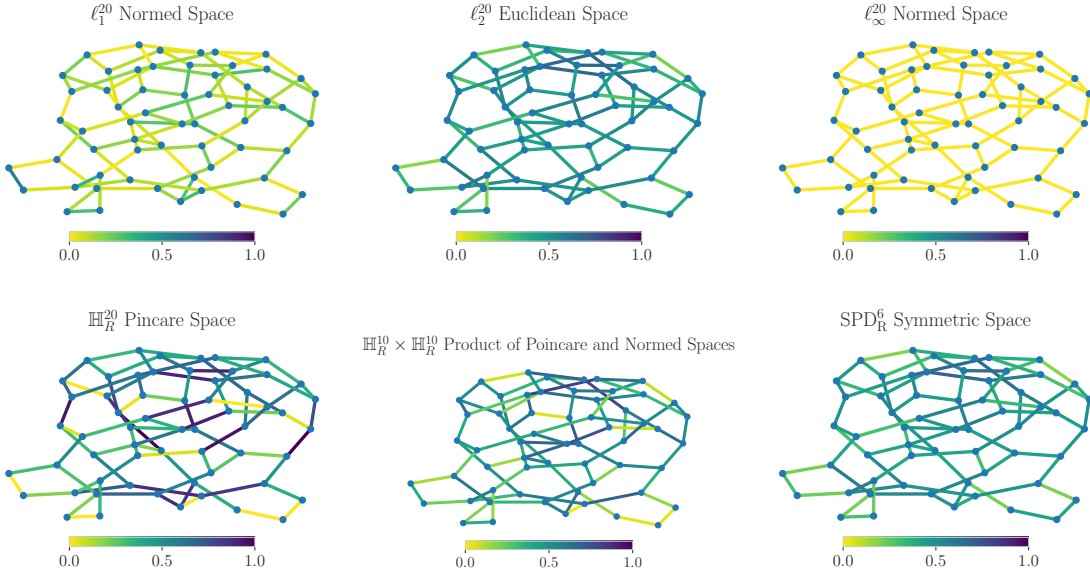

Figure 4: Embedding distortion shown in various spaces on a small expander-chordal graph. Color range indicates distortion levels.

otherwise $y_{u,v} = 0$. The BCE loss function is given by:

$$\mathcal{L} = \sum_{(u,v) \in \mathcal{T}} -y_{u,v} \cdot \log(\sigma(\phi_{f,b,X}(u,v))) - (1 - y_{u,v}) \cdot \log(1 - \sigma(\phi_{f,b,X}(u,v))), \tag{8}$$

where $\sigma(x) = \frac{1}{1+e^{-x}}$ is the sigmoid function. For each user $u$, we generate a negative set by randomly selecting one item that the user has not interacted with.

## E  Supplementary Graph Reconstruction Analysis

**Results of Expander Graphs.**   For readability, we choose to embed a small expander graph into various spaces and visually compare embedding distortion in these spaces, as displayed in Figure 4. We find that both normed spaces perform much better than other spaces. Further, we see that the graph undergoes small distortion in the $\ell_1$ space and unnoticeable distortion in the $\ell_\infty$ space when dealing with a small expander, although embedding expanders into normed spaces is a well-known challenge (Linial et al., 1995, Proposition 4.2).

## F  Trees, Grids, and Fullerenes

In our Large Graph Representational Capacity experiments (Section 4), we used trees, grids, and fullerenes as discretizations of manifolds with negative, zero, and positive curvatures, respectively. Refer to Figure 5 for a visual illustration of these discretizations. In chemistry, a *fullerene* is any molecule composed entirely of carbon in the form of a hollow spherical, ellipsoidal, or cylindrical mesh. In our experiments, we used combinatorial

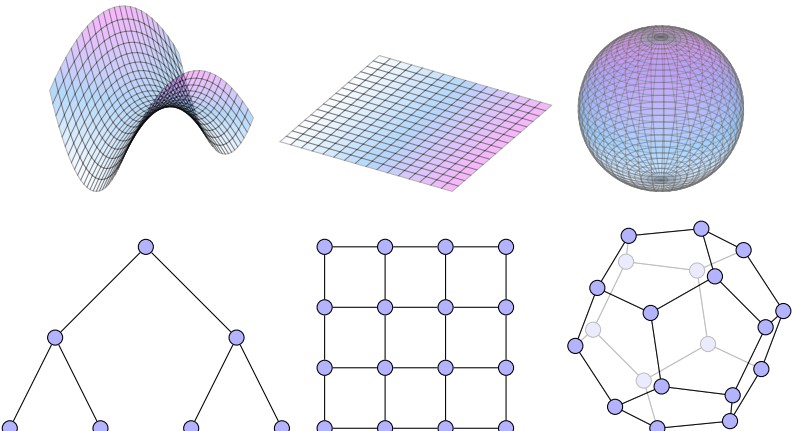

Figure 5: Top: surfaces of negative, zero, and positive curvature (from left to right). Bottom: graphs of negative, zero, and positive curvature (from left to right).

graphs representing spherical fullerenes. We generated the fullerene graphs using the `graphs.fullerenes()` function from SageMath (The Sage Developers, 2023). The number of possible fullerenes grows fast as a function in the number of nodes (OEIS Foundation Inc., 2023, A007894), and we used the first fullerene graph generated by `graphs.fullerenes()` for each node count. The graph data for the specific fullerenes used in our experiments can be found in our code repository, ensuring reproducibility and facilitating further analysis. Trees and grids are well-known and require no further description.

