# OpenReview forum: "Normed Spaces for Graph Embedding"
_TMLR — Accepted by TMLR_

### Review · Reviewer_rchM · 2024-01-28

**Summary Of Contributions:**

This work proposes to embed nodes of a graph in normed spaces. The authors use a very simple algorithm which randomly initializes the embeddings of the nodes and uses gradient descent to minimize an asymmetric loss function which compares norm distances against shortest path distances. The proposed embedding method is evaluated in the task of graph reconstruction, but also in the task of link prediction and in a recommendation task. In most cases, the normed space embeddings outperform other baseline embeddings.

**Audience:**

Yes

**Broader Impact Concerns:**

no concerns.

**Claims And Evidence:**

Yes

**Requested Changes:**

The requested changes listed below are related to the weaknesses of the paper presented above.

- From Section 3: every tree $T$ with $n$ vertices can be embedded isometrically in $\ell_\infty^{\mathcal{O}(\log n)}$. Will the proposed learning algorithm embed isometrically a tree $T$ with $n$ vertices in $\ell_\infty^{\mathcal{O}(\log n)}$? If the answer is no, please explain why.

- Give a justification why the proposed loss function is chosen over standard loss functions such as the mean squared error. I would suggest the authors provide some empirical results where the proposed function is replaced by the mean squared error.

- I would suggest the authors investigate whether the proposed embeddings could be used along with standard learning algorithms. For example, the authors could evaluate the learned embeddings + logistic regression in a node classification task (such as Cora, etc.).

- The authors should experiment with different embedding dimension sizes. Ideally, the embedding dimension size could be treated as a hyperparameter and thus be tuned.

- Please, provide some intuition on how a practitioner can choose which norm to employ in a specific application.

**Strengths And Weaknesses:**

Strengths:

- The proposed method is simple and can be integrated into different techniques. Evaluating the loss function is computationally efficient in case the shortest paths are pre-computed. Thus, the proposed method can be applied to very large graphs.

- The presented empirical results are strong. Normed space embeddings lead to low distortions and can thus tackle the graph reconstruction problem better than other types of embeddings.

- The paper is well-written and easy to read. All details are clearly explained.

Weaknesses:

- It is not clear to me how Section 3 is connected to the rest of the paper. In my understanding, the theoretical statements presented in this Section do not apply to the proposed embeddings and the authors should explain why.

- There is a lack of explanations or insights about the employed loss function. What are the advantages of such an asymmetric loss function over standard loss functions such as the mean squared error or the mean absolute error?

- The learned embeddings are not directly compatible with standard machine learning algorithms that operate in Euclidean spaces. Embeddings serve as general representations of objects that can be used in different downstream tasks. Could the learned embeddings be combined with a standard classification algorithm such as logistic regression or an MLP?

- In the reported results, the embedding dimension is always set to 20. Typically, the embedding dimension is treated as a hyperparameter since different values might lead to very different results.

- No intuition is given on why a specific norm ($\ell_\infty$) works better than the rest of the norms in most cases. It would be useful to a practitioner to know which norm to use in a specific application.

---

> ### Author Response · Authors · 2024-03-12
> **Response to Reviewer rchM**
>
> We thank Reviewer rchM for their thorough review. Below, we provide responses to the requested changes.
>
> 1. **Isometric embeddings of trees**: We emphasize that though the empirical results are strong, the surveyed theoretical results do not immediately translate to or predict the empirical results. As such, one could empirically embed many trees in low dimensions, but one would not be able to guarantee that that would be the case for all trees. We adjusted the language of the relevant paragraph in Section 3 to further emphasize that point. These results are surveyed to theoretically motivate normed spaces as embedding spaces, the same as one would provide similar theoretical motivations for hyperbolic geometry as an embedding space for hierarchical data.
>
> 2. **Standard mean squared error loss**: We opted for a loss function that minimizes distortion. We ran an empirical comparison with the mean squared error, as requested, and reported the results in Table 11 of the current revision. The results demonstrate that with respect to the Davg and mAP evaluation metrics, $\ell_\infty$ embeddings trained with both loss functions consistently outperform the other embeddings, and for each space, the embeddings trained with distortion loss outperform the embeddings trained with mean squared error. These results justify our choice.
>
> 3. **Node classification**: Could Reviewer rchM please elaborate? In Applications 1 and 2 from Sections 4.2 and 4.3, we demonstrated how the normed spaces and other embeddings can be integrated with geometric extensions of standard data-driven machine learning algorithms, which is standard practice in geometric machine learning, and at the heart of the field. For instance, we extend the implementation of Poincaré GCN from [CHA] to operate in product spaces, and the node embeddings are trained by optimizing the standard cross-entropy loss function. For node classification, there are principled extensions of logistic regression to metric spaces, e.g., the metric centroid-based layer NC-MM from [ZHA], that one could pair with our embeddings.
>
> 4. **Embedding dimension sizes**: We ran a further empirical analysis of the impact of embedding dimension on performance, and reported the new results in Tables 12-15 (in addition to Table  4 from the previous version). We note that for $\ell_1$ and $\ell_\infty$, performance generally improves as dimension grows bigger. We also note that increasing dimension does not seem to help much when there is a mismatch between the geometry of a space and a graph, such as is the case with $\ell_2^n$ and $H_R^n$ on Grid, and $H_R^n$ on Fullerenes. Lastly, we highlight that Lopez et al. [LOP] similarly evaluated embeddings of BIO-DISEASOME in Euclidean, hyperbolic and spherical spaces, their products, and Siegel space at dimension $n = 306$, and our low-dimensional normed spaced embedding outperformed all those embeddings. We mention Lopez’s results under “Space Dimension” in Section 4.1.
>
> 5. **Choice of norm**: The choice of the norm could be considered a hyperparameter to be tuned. In our experiments, $\ell_\infty$ performs best in the graph reconstruction task on 10/13 datasets, whereas for downstream tasks (link prediction and recommender systems), $\ell_1$ performs consistently best in almost all cases on five datasets. So, we recommend using $\ell_\infty$ for graph reconstruction and $\ell_1$ for downstream tasks. Alternatively, we recommend the product of $\ell_1$ and $\ell_\infty$ in all setups, as that product performs consistently as the second best option behind $\ell_1$ and $\ell_\infty$. We added this recommendation to Appendix C.1.
> We hope our responses and revisions address Reviewer rchM’s concerns and requests. We welcome any further comments or suggestions.
>
> References
>
> [CHA] Chami et al., 2019, Hyperbolic Graph Convolutional Neural Networks, NeurIPS 2019.
>
> [LOP] Lopez et al., 2021, Symmetric Spaces for Graph Embeddings: A Finsler-Riemannian Approach, ICML 2021.
>
> [ZHA] Zhao et al., 2023, Graph Neural Networks in Symmetric Positive Definite Matrices, ECML 2023.

---

> > ### Comment · Reviewer_rchM · 2024-03-19
> >
> > I appreciate the authors' responses and the updates in the manuscripts according to my comments.
> >
> > With regards to point 3, it is not clear to me how exactly the embeddings are learned in applications 1 and 2. If I am not wrong, both applications are formulated as classification problems, and then the embeddings are learned by minimizing the cross-entropy loss function. These embeddings are not general and depend on the task at hand. Instead,cI would expect the authors to train the model by minimizing Equation 1 (i.e., the distance-based loss function) and then, once the embeddings are learned to use some standard classifier such as logistic regression to make predictions for the two applications. In my view, this would provide a more clear picture of the effectiveness of the different embedding methods and the quality of the generated embeddings.

---

> > > ### Author Response · Authors · 2024-03-26
> > > **Follow-Up Response to Reviewer rchM**
> > >
> > > We thank Reviewer rchM for the careful clarification, valuable comments, and insightful suggestions, which have significantly improved our manuscript. Below, we address Point 3.
> > >
> > > > With regards to point 3, it is not clear to me how exactly the embeddings are learned in applications 1 and 2.
> > >
> > > In our work, the embeddings in the graph reconstruction (§4.1 - Benchmark), link prediction (§4.2 - Application 1), and recommender systems (§4.3 - Application 2) tasks are learned using a metric learning approach. Specifically, a learnable function embeds data in a metric space, and distances between data points are fed as features into a machine learning component. Then, the parameters of the embedding function and the machine learning component are learned jointly by minimizing a loss function using gradient descent. Motivated by the Reviewer’s question, we added a concise discussion on metric learning in Appendix C and summarized the components of the three metric learning pipelines in our work in Table 9.
> > >
> > > > These embeddings are not general and depend on the task at hand.
> > >
> > > We acknowledge and agree with the Reviewer's observation regarding the task-specific nature of the embedding learning process. Indeed, this characterizes metric learning and is a common thread in geometric machine learning. (We compare our work with, for instance, [GU] and [LOP], where the shortest path and task-specific metrics are separately considered in Sections 4.1 and 4.2, and Sections 5 and 7, respectively.) With that in mind, we emphasize that our work aims to empirically demonstrate that normed spaces perform very well as embedding spaces for a wide range of distance metrics, including shortest-path distance metrics of many families of graphs and distance metrics optimized for different tasks, in the context of geometric and metric machine learning.
> > >
> > > > Instead,cI would expect the authors to train the model by minimizing Equation 1 (i.e., the distance-based loss function) and then, once the embeddings are learned to use some standard classifier such as logistic regression to make predictions for the two applications.
> > >
> > > In response to this request, we carried out an additional experiment and reported the results in Appendix D1 and Table 17 in the current revision. In summary, while $\ell_1$ shortest-path metric embeddings achieve non-trivial accuracy on link prediction and outperform the corresponding $\ell_2$ and $\ell_\infty$ embeddings, they are still significantly outperformed by normed space embeddings learned with GNNs on link prediction. Although low-distortion metric embeddings have useful applications and are an important benchmark in geometric machine learning, they do not capture nuanced node features and high-order proximity. (In Section 4.1 under Motivation, we list examples of applications of low-distortion metric embeddings in approximation algorithms [LIN], online algorithms [BAN], and distributed algorithms [KHA].) Among the alternatives we examined in our work, we recommend $\ell_1$ metric embeddings learned with GNNs for link prediction.
> > >
> > > We welcome any further clarifications, comments, or suggestions, and thank the Reviewer in advance.
> > >
> > > References
> > >
> > > [BAN] Bansal et al., 2015, A polylogarithmic-competitive algorithm for the k-server problem, Journal of the ACM (JACM) 62.
> > >
> > > [GU] Gu et al., 2018, Learning mixed-curvature representations in product spaces, ICLR 2018.
> > >
> > > [KHA] Khan et al., 2008, Efficient distributed approximation algorithms via probabilistic tree embeddings, Proceedings of the twenty-seventh ACM symposium on Principles of distributed computing.
> > >
> > > [LIN] Linial et al., 1995, The geometry of graphs and some of its algorithmic applications, Combinatorica 15.
> > >
> > > [LOP] Lopez et al., 2021, Symmetric spaces for graph embeddings: A finsler-riemannian approach, ICML 2021

---

### Review · Reviewer_o9MM · 2024-02-03

**Summary Of Contributions:**

The authors propose using normed spaces, with particular emphasis on $\mathbb R^n_{\ell^1}, \mathbb R^n_{\ell^\infty}$, to represent nodes in a graph. They contrast this with metric spaces such as hyperbolic or spherical embeddings which have been motivated by the fact that their metric may more naturally align to capturing distances in certain graphs, and have provable bounds on the distortion. The authors of this work show empirically that embeddings in $\mathbb R^n_{\ell^1}$ and $\mathbb R^n_{\ell^\infty}$ can be effectively trained via gradient descent to outperform a variety of alternative geometries, including hyperbolic and spherical, on graph reconstruction, link prediction, and recommendation.

**Audience:**

Yes

**Claims And Evidence:**

Yes

**Requested Changes:**

**Asymmetric Loss Function:** In the discussion here the authors claim that Figure 4 (b) and (c) shows that the asymmetry in the loss function has a bigger impact on the small tree. This is not clear to me from the figures. The asymmetry in the loss function would have suggested that nodes which are closer than they should be get less of a penalty than nodes which are farther apart, but the graphs show the opposite (namely, that smaller trees have a larger number of nodes which are farther apart than they should be). In addition, I think more graphs should be inspected if one wants to make a more general statement in this regard. For example, trying a variety of graph sizes and calculating, for each one, the number of pairs which over vs. under-estimate the distance, and then plotting the average of this for each graph size. To be fair, however, I think the asymmetry is not much of an issue - after all, the loss function directly targets the distortion, so it is minimizing exactly the metric it will be evaluated on.

**Link Prediction:** In Section 4.1 "baseline experiments", the authors state that their focus was on settings with small dimensions, where it was known that geometric embeddings tended to outperform Euclidean. The results in Table 3 suggested $\mathbb R_{\ell^2}$ and $\mathbb R_{\ell^\infty}$ achieved lower distortion even when $n$ was small. When evaluating link prediction in Section 4.2, however, the authors switch to using a larger embedding dimension. Are larger embeddings required to see a benefit here? Do the results of Section 4.1, where $\mathbb R^n_{\ell^2}$ and $\mathbb R_{\ell^\infty}^n$ outperformed alternative geometric embeddings in low dimensions, not transfer to this setting?

**Training vs. Representational Capacity:** Some of the proposed baselines have very strong theoretical bounds on distortion for certain graphs - for example, the fact that hyperbolic space can embed any tree with arbitrarily low distortion - whereas the results for normed spaces are less impressive. Thus, it seems to me that an implication of the author's claims is that normed spaces can be *trained* more effectively than more complicated geometric counterparts. On the other hand, for $\mathbb R_{\ell^\infty}$ gradient descent would essentially be operating as coordinate descent, since only one dimension would appear at a time. Have I correctly inferred the implication of the author's claims, and if so how is this explained in light of the sparse gradient in the $\mathbb R_{\ell^\infty}$ setting?

**Strengths And Weaknesses:**

### Strengths
The paper is very well written, includes the relevant background and cites prior work appropriately. The authors support their claims in both synthetic settings as well as real-world tasks, and the results look significant. The author's proposed embedding space is also computationally more efficient than the baselines, and more straightforward to implement.

### Weaknesses
No theoretical justification for the observed results are presented. The author's claims rest entirely on empirical observation, and since all methods under consideration are trained via gradient descent this leaves open the possibility that hyperparameter tuning or different training techniques could allow opposite results to be achieved. The size of the graphs under consideration are relatively small, it is unclear if these benefits would translate to graphs of larger scale.

---

> ### Author Response · Authors · 2024-03-12
> **Response to Reviewer o9MM**
>
> We thank Reviewer o9MM for their thoughtful review. Below, we provide responses to the requested changes.
>
> 1. **Asymmetric loss function**: We second the reviewer’s opinion that reaching an empirical conclusion on the asymmetry of the loss function would require investigating more graphs at different sizes, and also that asymmetry of the loss function is not much of an issue. (We are considering removing that part from the final version.) We wanted to clarify that whereas the amount by which two nodes could be closer than they should is bounded (since distances cannot go below 0), the amount by which they could be further apart could potentially be unbounded, but that seems to not be an issue in practice, as embeddings are observed to stay within a narrow window of the correct distances.
>
> 2. **Link prediction**: We report the evaluation results for dimensions 32, 64, and 128 in Table 14. We note that in our experiments, $\ell_1$ performs consistently best across GNNs and dimensions from 32 to 128, and that overall, the results improve with increased dimension.
>
> 3. **Training vs. representational capacity**: Could the reviewer please elaborate on the kind of thought-for explanation for the effectiveness of $\ell^\infty$ in light of the sparse gradient? (E.g., does it relate in this context to the running times from Figure 3?) Motivated by the reviewer’s note, we clarify our claim that “normed spaces are empirically observed to be easier to train and to perform better in general graph embedding settings that leverage gradient descent” in Section 1. We now also mention under Limitations in Section 5 that “it is possible that hyperparameter tuning or different spaces and training techniques than normed spaces and gradient descent can achieve better performance in general settings, in addition to the restricted cases where some geometric spaces perform exceptionally well on particular graphs.”
>
> Lastly, we added more emphasis on the importance of analyzing how the benefits would translate to graphs of larger scale to Limitations in Section 5.
>
> We hope our responses and intended revisions address Reviewer o9MM’s concerns and requests. We welcome any further comments or suggestions.

---

### Review · Reviewer_oocY · 2024-02-27

**Summary Of Contributions:**

This paper explores the normed spaces to embed finite metric spaces with low distortion bounds in low dimensions. Drawing inspiration from the theoretical foundations in prior works (Johnson & Lindenstrauss, 1984; Bourgain, 1985; Linial et al., 1995;  Krauthgamer et al., 2004 ), the authors advocate for normed spaces as a flexible and computationally efficient alternative to popular Riemannian manifolds for learning graph embeddings. Through extensive experimentation on synthetic and real-world graph reconstruction benchmarks, normed space embeddings demonstrate superior performance compared to traditional manifolds while requiring significantly fewer computational resources. Additionally, empirical validation across graph families associated with various curvatures underscores the adaptability of normed spaces in capturing diverse graph structures as graph sizes increase. Furthermore, the paper showcases the practical utility of normed space embeddings in tasks such as link prediction and recommender systems, offering a valuable tool for geometric graph representation learning.

**Audience:**

Yes

**Claims And Evidence:**

Yes

**Requested Changes:**

The paper is generally well written, however, from the point of view of a general reader in ML it would be helpful to make these changes.

1. Section 3 would be much better with some more background on normed spaces and other embedding spaces. I can see it in the Appendix, it would be helpful to have abridged version of it in the main paper.

2. In Section 3. the main point being made is "large classes of finite metric spaces can in theory be abstractly embedded with low theoretical bounds on distortion in low dimensional normed spaces". It would be better to clearly state this at the start the section  and then give the evidence by discussing the prior works as it is done in the middle paragraphs. If possible add some more details about the the kind of metric spaces for which the results apply. It would be great if these results can be discussed in depth in the Appendix and have a table summarizing the results in the main paper.

3. The authors did not discuss pseudoEuclidean embeddings (PSE). These are also a great choice for embedding discrete metric spaces since any discrete metric space is isometrically embeddable in PSE spaces and they also contain normed vector spaces. See figure 2.1 in [1].  I am curious to see how these embeddings will perform for the settings considered in the paper. Please see the references below for PSE.

4. Have a table (maybe in the appendix) summarizing various embedding spaces used in the experimental results.

[1] Goldfarb 1985, A new approach to Pattern Recognition,  Progress in Pattern Recognition.

[2] Vishwakarma & Sala, 2022, Lifting Weak Supervision to Structured Prediction, NeurIPS 2022.

**Strengths And Weaknesses:**

Strengths:

1. The paper offers a novel approach by underscoring normed spaces as a more flexible, computationally efficient, and technically less challenging alternative to popular methods for learning graph embeddings.

2. The proposal is grounded in theoretical results from discrete geometry, providing a solid foundation for using normed spaces in embedding finite metric spaces with low distortion bounds in low dimensions.

3. Empirical evaluation on synthetic and real-world benchmark graph datasets demonstrates the superior representational capacity of normed spaces, outperforming various Riemannian manifold alternatives across diverse graph structures and curvatures.


Weaknesses:
I do not see any major weaknesses. The requested changes section lists my concerns, questions, and suggestions below.

---

> ### Author Response · Authors · 2024-03-12
> **Response to Reviewer oocY**
>
> We thank Reviewer oocY for their supportive review. Below, we provide responses to the requested changes.
> 1. **Section 3 and the appendix**: As requested, we added two tables in the appendix, which summarize the theoretical results and embedding spaces investigated in this work. We provide links to these tables at the beginning of Section 3.
> 2. **Pseudo-Euclidean embeddings**: We embedded the synthetic graphs from our paper in the pseudo-Euclidean space $\mathbb{R}^{10,10}$, and reported the results in Table 1 of the current revised version. We note that though PSE can outperform $\ell_2$ in our setting, it is still outperformed by $\ell_1$ and $\ell_\infty$.
>
> We hope our responses and intended revisions address Reviewer oocY’s concerns and requests. We welcome any further comments or suggestions.

---

### Decision · Action_Editor_FMKp · 2024-04-28

**Recommendation:** Accept as is

**Comment:**

The paper studies network embeddings into normed spaces. This builds on earlier work embedding graphs into Riemannian manifolds, which showed that Euclidean embeddings can be outperformed (i.e., lower distortion at the same number of dimensions). The authors propose an even simpler approach. The work presents empirical results showing very strong results.

All reviewers suggest accepting. I agree with their assessment. This is strong empirical work showing how a simple idea works very well. Such work is very useful, especially for a field where the trend is to look for more and more exotic manifolds to embed into.

**Audience:**

Yes, obtaining high-quality representations are an important area of machine learning, and this paper fits nicely into it.

**Claims And Evidence:**

Yes, the paper has strong experimental support for its claims.